

**Rapid mass growth and enhanced light extinction of atmospheric aerosols during the heating**
**season haze episodes in Beijing revealed by aerosol-chemistry-radiation-boundary layer**
**interaction**
Zhuohui Lin[1], Yonghong Wang[2], Feixue Zheng[1],Ying Zhou[1],Yishuo Guo[1], Zemin Feng[1], Chang Li[1],
Yusheng Zhang[1], Simo Hakala[2], Tommy Chan[2], Chao Yan[2], Kaspar R. Daellenbach[2], Biwu Chu[3],
Lubna Dada[2], Juha Kangasluoma[1,2], Lei Yao[2], Xiaolong Fan[1], Wei Du[2], Jing Cai[2], Runlong Cai[2], Tom
V. Kokkonen[2,4], Putian Zhou[2], Lili Wang[5], Tuukka Petäjä[2,4], Federico Bianchi[1,2],Veli-Matti
Kerminen[2,4],Yongchun Liu[1], and Markku Kulmala[1,2,4]
[1]Aerosol and Haze Laboratory, Beijing Advanced Innovation Center for Soft Matter Science and
Engineering, Beijing University of Chemical Technology, Beijing, China
[2]Institute for Atmospheric and Earth System Research / Physics, Faculty of Science, University of
Helsinki, Finland
[3]Research Center for Eco-Environmental Sciences, Chinese Academy of Science, Beijing, China
[4]Joint international research Laboratory of Atmospheric and Earth SysTem sciences (JirLATEST),
Nanjing University, Nanjing, China
[5]State Key Laboratory of Atmospheric Boundary Layer Physics and Atmospheric Chemistry (LAPC),
Institute of Atmospheric Physics, Chinese Academy of Sciences, Beijing 100029, China
Corresponding author: Yonghong Wang
E-mail: yonghong.wang@helsinki.fi
Submitted to: Atmospheric Chemistry and Physics





**Abstract**
Despite the numerous studies investigating haze formation mechanism in China, it is still puzzling
that intensive haze episodes could form within hours directly following relatively clean periods. Haze
has been suggested to be initiated by the variation of meteorological parameters and then to be
substantially enhanced by aerosol-radiation-boundary layer feedback. However, knowledge on the
detailed chemical processes and the driving factors for extensive aerosol mass accumulation during
the feedback is still scarce. Here, the dependency of the aerosol number size distribution, mass
concentration and chemical composition on the daytime mixing layer height (MLH) in urban Beijing
is investigated. The size distribution and chemical composition-resolved dry aerosol light extinction
is also explored. The results indicate that the aerosol mass concentration and fraction of nitrate
increased dramatically when the MLH decreased from high to low conditions, corresponding to
relatively clean and polluted conditions, respectively. Particles having their dry diameters in the size
of ~400-700 nm, and especially particle-phase ammonium nitrate and liquid water, contributed
greatly to visibility degradation during the winter haze periods. The dependency of aerosol
composition on the MLH revealed that ammonium nitrate and aerosol water content increased the
most during low MLH conditions, which may have further triggered enhanced formation of sulphate
and organic aerosol via heterogeneous reactions. As a result, more sulphate, nitrate and water soluble
organics were formed, leading to an enhanced water uptake ability and increased light extinction by
the aerosols. The results of this study contribute towards a more detailed understanding of the aerosol-
chemistry-radiation-boundary layer feedback that is likely to be responsible for explosive aerosol
mass growth events in urban Beijing.










## 1. Introduction

Despite the recent reduction of air pollutants and their precursors in China between 2013 and 2017, the current emission and air pollution levels are still substantially high (Wang et al., 2020b; Zheng et al., 2018). Such high emissions, combined with specific meteorological conditions, frequently lead to severe haze episodes (An et al., 2019; Wang et al., 2019). Particulate matter, a major air pollutant, has considerable effects on climate, human health and visibility degradation (Che et al., 2007; Lelieveld et al., 2015; Spracklen et al., 2008; Wang et al., 2015).

During winter haze episodes, a rapid growth of the aerosol mass concentration has commonly been observed, and this phenomenon seems to be directly affected by meteorological factors (Li et al., 2018b; Liu et al., 2018, 2019b; Wang et al., 2018a, 2014a). The meteorological conditions and increased aerosol concentrations are proposed to be interlinked by a feedback loop, called the aerosol-chemistry-boundary layer feedback, in which aerosol particles reduce both solar radiation reaching the surface and turbulent kinetic energy of the near-surface air(Ding et al., 2016; Petäjä et al., 2016; Wang et al., 2020d). The increased stability of the boundary layer leads to enhanced air pollution in the mixed layer, which further suppresses the development of boundary layer. As a consequence, concentrations of primary aerosol particles, water vapor and relative humidity increase, creating more favourable conditions for homogeneous and heterogeneous on aerosol surfaces or inside them (Cheng et al., 2016a; Wang et al., 2016; Wu et al., 2018). Such reactions cause rapid formation of secondary aerosol matter and enhanced light extinction during severe winter haze episodes. However, more detailed information on the aerosol and reactive gas chemistry during the aerosol-chemistry-boundary layer feedback and related rapid aerosol mass growth events is still needed (Liu et al., 2019). For instance, it is still unclear which chemical reactions and which compounds in the particulate matter play key roles during such rapid mass growth events.

The particle number size distribution and chemical composition are considered to be the most important variables influencing the light extinction by aerosol particles. In the atmosphere, the highest contribution to aerosol light extinction comes from organic compounds, nitrate and sulphate in



particles with diameters of a few hundred nm. This is due to the dominant mass fractions of the
aforementioned compounds in aerosols that correspond to the peak intensity of solar radiation at
wavelengths around 550 nm (Jimenez et al., 2009; Swietlicki et al., 2008). In addition, light scattering
which contributes the most to the light extinction by atmospheric aerosols, can be substantially
enhanced by the presence of liquid water in the aerosol (Chen et al., 2014; Liu et al., 2019a; Pan et
al., 2009; Wang et al.,2020). Hence, quantifying the response of light extinction to different chemical
compounds would be helpful in evaluating the feedbacks associated with secondary aerosol
production.

In this study, we focus on the physical and chemical properties of aerosols in Beijing during the winter
heating season from October 2018 to February 2019 using state-of-the-art instrumentation. The
variation of aerosol chemical composition and the associated light extinction coefficient as a function
of the varying mixing layer height are discussed. Our aim is to identify the key chemical components
which contribute to the aerosol-chemistry-radiation-boundary layer feedback loop in Beijing.

**2.   Methodology**
**2.1. Measurement location and instrumentations**
Measurements were conducted between 1 October 2018 and 28 February 2019 at the roof top of the
university building at the west campus of Beijing University of Chemical Technology (39.95°N,
116.31°E). This station is located about 150 m away from the nearest road (Zizhuyuan road) and 500
m away from the West Third Ring Road, and it is surrounded by commercial properties and residential
dwellings representative of an urban environment. More details on the location can be found in (Liu
et al., 2020; Zhou et al., 2020).

The meteorological data for this work include basic meteorological variables (relative humidity (RH),
temperature, wind speed, wind direction, and visibility) and mixing layer height (MLH) measured


using a weather station (Vaisala Inc., Finland) and a Ceilometer CL51 (Vaisala Inc., Finland),
respectively. The MLH is defined as the height above the surface, through which relatively vigorous
vertical mixing occurs (Holzworth, 1972), and its value is highly related to the vertical temperature
structure and, so some extent, to a mechanically-induced turbulence (Baxter, 1991). Here, we
followed the method introduced earlier by Münkel et al. (2007) and  Eresmaa et al. (2012) in
determining the MLH.

The number concentration of clusters or small aerosol particles in the size range from 1.3-2.5 nm and
the number size distributions of aerosol particles from 6 nm to 840 nm were measured by a Particle
Sizer Magnifier (PSM) and a Differential Mobility Particle Sizer (DMPS), respectively (Aalto et al.,
2001; Vanhanen et al., 2011). The mass concentration of fine particulate matter ($PM_{2.5}$) was measured
using a Tapered Element Oscillating Microbalance Dichotomous Ambient Particulate Monitor
(TEOM 1405-DF, Thermo Fisher Scientific Inc, USA) with a total flow rate of 16.67 L/min (Wang
et al., 2014).

A time-of-flight aerosol chemical speciation monitor (ToF-ACSM, Aerodyne Research Inc.) was used
to measure the concentrations of non-refractory (NR) components, including sulfate, nitrate,
ammonium, chloride and organics of $PM_{2.5}$ (Fröhlich et al., 2013). The inlet flow was set at 1.4 $cm^3/s$.
The particle beam passed through the chamber and reached the heated porous tungsten surface
(T≈600℃). There, the non-refractory $PM_{2.5}$ constituents were vaporized and then ionized by electrons
($E_{kin}$=70eV, emitted by a tungsten filament). The ions were measured by a detector and the data was
analyzed using Tofware ver. 2.5.13 within IgorPro ver. 6.3.7.2 (WaveMetrics). The relative ionization
efficiencies (RIE) for sulfate, nitrate, ammonium, chloride and organics applied were 0.86, 1.05, 4.0,
1.5 and 1.4, respectively. In addition to the RIE corrections, $CO_2^+/ NO_3$ artifact correction and
collection efficiency (CE) correction (Pieber et al., 2016) were also applied to the data. Mass
concentrations of ammonium nitrate, ammonium sulfate and ammonium chloride were determined
according to the method introduced by Gysel et al. (2007). The aerosol liquid water content (AWC)
was calculated by thermodynamic equilibrium model ISORROPIA II using ToF-ACSM data



(Fountoukis and Nenes, 2007).

Highly-oxygenated organic molecules (HOMs) were measured by a chemical ionization long time-
of-flight mass spectrometer equipped with a nitrate chemical ionization source (LToF-CIMS,
Aerodyne Research, Inc. USA) (Jokinen et al., 2012) similar to gas-phase sulfuric acid. Organic
carbon (OC) and element carbon (EC) concentrations were measured semi-continuously with a 1-
hour time resolution using an OC/EC Analyzer (Model-4, Sunset Lab. Inc.).

The air mass history was studied by calculating particle retroplumes using a Lagrangian particle
dispersion model FLEXPART (FLEXible PARTicle dispersion model) ver. 9.02 (Stohl et al., 2005).
The ECMWF (European Centre for Medium-Range Weather Forecast) operational forecast (with 0.15°
horizontal and 1 h temporal resolution) was used as the meteorological input into the model. During
the measurement period, a new release of 50 000 test particles, distributed evenly between 0 and 100
m above the measurement site, occurred every 1 hour. The released particles were traced backwards
in time for 72 h, unless they exceeded the model boundary (20–60°N, 95–135°E).

**2.2.  Aerosol light extinction calculation**
The aerosol light extinction coefficient was calculated with the Mie-Model, which uses particle
number size distribution, mass concentrations of different aerosol compounds and their refractive
index as inputs (Seinfeld and Pandis, 2006). We introduced a series of assumptions into the Mie-
Model, including 1) "internal mixture" which considers each chemical component in a particle as
homogeneously mixed with each other; 2) all particles are spherical; and 3) particles of different sizes
have the same chemical composition.

The practical method introduced under those assumptions in previous studies were found to be
capable of estimating a variation trend of optical property of $PM_{0.5-20}$ with a relatively good accuracy
(Lin et al., 2013).





Table 1. Summary of the parameters for calculating the average optical refractive index.

| Specie | $\rho_i(\mathrm{g\,cm^{-3}})$ | $n_i$ | $k_i$ |
|---|---|---|---|
| (NH₄)₂SO₄ | 1.760 | 1.530 | 0.000 |
| NH₄NO₃ | 1.725 | 1.554 | 0.000 |
| NH₄Cl | 1.527 | 1.639 | 0.000 |
| Organics | 1.400 | 1.550 | 0.001 |
| EC | 1.500 | 1.800 | 0.540 |


The average optical refractive index (AORI) of an internally-mixed particle can be calculated from
the optical refractive indices (ORI) of each chemical component by following a mixing rule of
volume-averaged chemical components as AORI = $n_{\mathrm{eff}}$ + $k_{\mathrm{eff}}$ × i, where the real part ($n_{\mathrm{eff}}$) and
imaginary part ($k_{\mathrm{eff}}$) are given by:

$$n_{eff} = \left(\sum_i n_i \cdot m_i/\rho_i\right)\bigg/\left(\sum_i m_i/\rho_i\right) \qquad (1)$$

$$k_{eff} = \left(\sum_i k_i \cdot m_i/\rho_i\right)\bigg/\left(\sum_i m_i/\rho_i\right) \qquad (2)$$

Here $m_i$ and $\rho_i$ are the mass concentration and density of the component $i$ in particles, respectively,
and $n_i$ and $k_i$ are the real and imaginary parts of ORI of this component, respectively. The
parameters for calculating the AORI are summarised in Table 1. The values of $n_i$ and $k_i$ in Table 1
are referenced to the light wavelength of 550 nm.

$Q_{sp,j}$ represents light scattering efficiency of a single particle with diameter $D_j$, while $Q_{ep,j}$
represents light absorption efficiency. Theoretically, $Q_{sp,j}$ and $Q_{ep,j}$ are both the function of $D_j$ and
the $AORI_j$ (the AORI of the particle with diameter $D_j$) at a given light wavelength λ, for which the
complicated calculations were referenced to a previous publication. Regarding the limitations of
measurement techniques, the $AORI_j$ was assumed to be equal to the AORI$_{PM2.5}$, which was




determined based on chemical composition of $PM_{2.5}$. It is possible to derive expressions for the cross
sections of a spherical particle exactly. The formulas for $Q_{sp,j}$ and $Q_{ep,j}$ are:

$$Q_{sp,j}(D_j, \lambda, AORI_j) = \frac{2}{\alpha^2} \sum_{k=1}^{\infty} (2k+1) \cdot [|a_k|^2 + |b_k|^2] \qquad (3)$$

$$Q_{ep,j}(D_j, \lambda, AORI_j) = \frac{2}{\alpha^2} \sum_{k=1}^{\infty} (2k+1) \cdot Re[a_k + b_k] \qquad (4)$$


where


$$a_k = \frac{\alpha\psi_k'(y)\psi_k(\alpha) - y\psi_k'(\alpha)\psi_k(y)}{\alpha\psi_k'(y)\xi_k(\alpha) - y\xi_k'(\alpha)\psi_k(y)}$$



$$b_k = \frac{y\psi_k'(y)\psi_k(\alpha) - \alpha\psi_k'(\alpha)\psi_k(y)}{y\psi_k'(y)\xi_k(\alpha) - \alpha\xi_k'(\alpha)\psi_k(y)}$$


with $y = \alpha m$.


$$m = n_{eff} + i \cdot k_{eff}$$



$$\alpha = \frac{\pi D_j}{\lambda}$$


with $\lambda$ = 550 nm.

where complex number m stands for $AORI_j$, while $\alpha$ is the size of the particle, usually expressed as
a dimensionless size parameter. The functions $\psi_k(z)$ and $\xi_k(z)$ are the Riccati–Bessel functions:

$$\psi_k(z) = \left(\frac{\pi z}{2}\right)^{1/2} J_{k+1/2}(z) \qquad (5)$$

$$\xi_k(z) = \left(\frac{\pi z}{2}\right)^{1/2} \left[J_{k+1/2}(z) + i(-1)^k J_{-k-1/2}(z)\right] \qquad (6)$$




where $J_{k+1/2}$ and $J_{-k-1/2}$ are the Bessel functions of the first kind and their footnotes indicate the
order of Bessel functions. The Mie theory can serve as the basis of a computational procedure to
calculate the scattering and absorption of light by any sphere as a function of wavelength.

According to the Mie-Model, $b_{sp}$ (light scattering coefficient) and $b_{ep}$ (light extinction coefficient)
can be quantified with Eqs. (5) and (6), respectively. $b_{ap}$ (light absorption coefficient) is the
difference between $b_{ep}$ and $b_{sp}$, which equals zero, when $k_i$ equals zero or very small. Optical
properties including $b_{ep}$, $b_{sp}$ and $b_{ap}$ to be discussed later are all referenced to light wavelength of
550 nm.

$$b_{sp} = \sum_j b_{sp,j} = \sum_j \frac{\pi D_j^2}{4} \cdot Q_{sp,j}(D_j, \lambda, AORI_j) \cdot N_j \qquad (7)$$

$$b_{ep} = \sum_j b_{ep,j} = \sum_j \frac{\pi D_j^2}{4} \cdot Q_{ep,j}(D_j, \lambda, AORI_j) \cdot N_j \qquad (8)$$



In Eqs. (7) and (8), $D_j$ stands for the median Stokes diameter in the j-th particle size range and $N_j$ is
the number concentration of particles with diameter, $D_j$.

**3. Results and discussion**
**3.1. Typical case of rapid aerosol mass growth episodes affected by aerosol-chemistry-**
**boundary layer interactions**
An example of rapid aerosol mass growth in urban wintertime Beijing is illustrated in Figure 1, where
the haze accumulation was associated with a rapid $PM_{2.5}$ mass concentration increase from few $\mu g/m^3$
to more than 100 $\mu g/m^3$ in less than 7 hours. A haze episode started on afternoon 20 February 2019
under stagnant meteorological conditions with low wind speeds and elevated ambient relative
humidity (Figure S1). The polluted periods during this case occurred under southerly wind transport





conditions, whereas clean air masses originated from the north-westerly regions (as shown in Figure
S2, S3). These are typical features for a haze evolution process in Beijing (Wang et al., 2020b). During
the haze periods marked by the shaded areas in Figure 1, an obvious increase of chemical mass
concentration was observed by the ToF-ACSM, characterised by high concentrations of secondary
aerosol components (nitrate, organics and sulphate) and typically a shallow boundary layer. The mass
concentrations of organics, sulphate and nitrate increased dramatically with a decreasing MLH,
accounting for 88.5% of NR-PM$_{2.5}$ during the rapid aerosol mass growth period. The aerosol mass
growth was the fastest for nitrate. The mass concentrations of organic and elemental carbon followed
that of NR-PM$_{2.5}$.

The MLH reached its maximum at around 14:00 in the afternoon of 20 February, after which the
development of the mixing layer was suppressed and MLH decreased with the arrival of pollution
(Figure 1a). Previous studies have shown that the aerosol-radiation-boundary layer feedback
contributes to a rapid enhancement of air pollution (Petäjä et al., 2016; Wang et al., 2020d). High
concentrations of aerosol particles obscure downward radiation, as a result of which the surface
temperature and sensitive heat flux decrease and the development of mixing layer height is suppressed.
Recent studies have gradually realized that the facilitation of various chemical processes play a non-
negligible role in the aerosol-radiation-boundary layer feedback (Liu.Q et al., 2018; Liu. Z et al.,2019).
Therefore, it is important to identify and quantify the role of different specific chemical species and
particle size ranges in reducing atmospheric radiation and extinction.

Figure 2 shows the contributions of size and chemical composition-resolved dry aerosol to light
extinction during the investigated period. As the pollution intensified and MLH decreased (Fig 1c),
the light extinction of atmospheric aerosols increased significantly. Assuming that particles of
different sizes have the same chemical composition as PM$_{2.5}$ (organics, NH$_4$NO$_3$, EC, (NH$_4$)$_2$SO$_4$,
NH$_4$Cl), the light extinction of particles in the size range of 300-700 nm increased significantly from
the relative clean period to the polluted period (namely from 12:00 to 16:00). During relatively clean
conditions, the contributions of organics, NH$_4$NO$_3$, EC, (NH$_4$)$_2$SO$_4$ and NH$_4$Cl to the total aerosol



light extinction were 42%, 23%, 18%, 11% and 7%, respectively. The contribution of $NH_4NO_3$ to
aerosol light extinction reached 40% during the heavily polluted period. The increased light extinction
by aerosols reduced solar radiation reaching the surface, so that the development of the boundary
layer was suppressed.

**3.2. Connection between the aerosol chemical composition, light extinction, size distribution**

259        **and MLH during the heating season**


To better characterize the effect of the chemical composition of dry aerosols and the PNSD (particle
number size distribution) light extinction under different MLH conditions, the daytime (8:00 – 16:00
LT) measurement data from October 2018 to February 2019 were selected for further analysis. As
shown by Figure 3 and consistent with other observations in Beijing (Tang et al., 2016; Wang et al.,
2020c), there was a general tendency for the $PM_{2.5}$ mass concentration to increase with a decreasing
MLH. Organic compounds and nitrate were the most abundant fractions of the daytime aerosol mass
composition, contributing together approximately 70% to total $NR-PM_{2.5}$ mass concentration. With a
decreasing MLH, the fraction of nitrate mass in $NR-PM_{2.5}$ slightly increased while that of organics
decreased. This feature makes the aerosol more hygroscopic under low MLH conditions typical for
heavily polluted periods. The increased nitrate fraction in the aerosol could also enhance the
formation of other secondary aerosol components (Xue et al., 2019). Note that some fraction of
aerosol nitrate could consist of organic nitrate originating from reaction of peroxy radical with nitric
oxide; however, it is difficult to distinguish organic nitrate from inorganic nitrate at the moment due
to instrumental limitations (Fröhlich et al., 2013).

Figure 4 depicts the calculated daytime light extinction of the dry aerosol as a function of the MLH,
separated by different size ranges and chemical components. We may see that in general, particles
with dry diameters in the range of 300-700 nm explained more than 80% of the total aerosol light
extinction (Figure 4b). Similar to their share in $NR-PM_{2.5}$, the fraction of light extinction by





ammonium nitrate increased and that of organics decreased during the lowest MLH conditions
corresponding to the heavy pollution periods (Figure 4d). There are also apparent differences in the
relative contribution of different particle size ranges to light extinction in different MLH conditions:
with a decreasing MLH, the contribution of particles with dry dimeters larger than about 400-500 nm
clearly increased while that of sub-300 nm particles notably decreased. This indicates that the
enhanced light extinction by the dry aerosol at low MLH conditions was not only due the more
abundant aerosol mass concentration, but also due to the growth of individual particles to optically
more active sizes.

At relative humidity larger than about 70%, aerosol liquid water gives a significant contribution to
the aerosol mass concentration and often a dominant contribution to the aerosol light extinction (Titos
et al., 2016). This has important implications for the aerosol-chemistry-radiation-boundary layer
feedback, when considering our findings listed above and further noting that heavy pollution periods
are often accompanied by high values of RH in Beijing (Zhong et al., 2018). First, compared to clean
or moderately-polluted conditions, the enhancement in the aerosol light extinction under polluted is
probably much larger than that illustrated in Figure 4. Second, the high aerosol water content under
polluted conditions promotes many kinds of chemical reactions taking place on the surface or inside
aerosol particles.

**3.3. Aerosol-chemistry-radiation-boundary layer interaction**

In order to further investigate the interaction between MLH and chemical compounds (either observed
or calculated), we divided the observed $PM_{2.5}$ concentrations into highly polluted and less polluted
conditions using a threshold value of 75 $\mu g\ m^{-3}$ for $PM_{2.5}$. The organics, nitrate, ammonium, sulfate,
chloride, HOM, aerosol water content (AWC) and $PM_{2.5}$ as a function of the mixing layer height
during both highly polluted and less polluted conditions are shown in Figure 5. The fitted relationships
connecting the concentrations of different chemical compounds to the reduction of MLH under highly



and less polluted conditions allowed us to estimate the net mass concentration increase of each
compound due to secondary formation and aerosol-chemical-boundary layer feedback under highly
polluted conditions (shaded areas in Figure 5). It is worth noting that AWC, nitrate and sulfate
increased the most as the MLH decreased, as represented by the large shaded areas in Figs. 5 (h), (b)
and (c). The day-time nitrate in aerosol is formed predominately via the reaction of nitric acid and
ammonium, while nitric acid is produced from gas phase reaction of nitrogen dioxide and hydroxy
radical (Seinfeld and Pandis, 2006). High concentrations of daytime nitrate aerosols indicate efficient
production of gas phase nitric acid, its partitioning into liquid aerosol and its fast neutralization by
abundant ammonia (Li et al., 2018a; Pan et al., 2016; Wang et al., 2020). A recent study shows that
condensation of nitric acid and ammonia could promote fast growth of newly formed particle in urban
environment condition (Wang et al., 2020d). Another possibility is that ammonium nitrate is formed
rapidly on particle surfaces due to the hydrolysis of dinitrogen pentoxide ($N_2O_5$) during daytime, as
the AWC increased significantly (Wang et al., 2014;Wang et al.,2020). However, a quantitative
distinction between the two formation pathways for nitrate formation is not possible in this study. The
dramatic increase of nitrate aerosol could also promote the formation of sulfate by heterogeneous
reactions (Cheng et al., 2016b; Wang et al., 2016). The concentration of HOMs showed a slight
increase as the MLH decreased, which suggests that also the formation of HOMs is enhanced with an
increased level of air pollution. This phenomenon should be further investigated as HOMs can
substantially contribute to the secondary organic aerosol formation.

Figure 6 displays the dry aerosol light extinction by different chemical compounds in the same way
as Fig. 5 did for aerosol mass concentrations. The aerosol light extinction is directly related to the
reduction of solar radiation reaching the surface, assuming that aerosol chemical components are
vertically nearly homogeneously distributed. The light extinction from ammonium nitrate,
ammonium sulfate and organics showed significantly increased contributions under highly polluted
conditions (low MLH) as compared with less polluted conditions. To the contrary, no such
enhancement was observed for ammonium chloride or element carbon (Figs. 6 (d) and (e)). In case
of EC this is an expected result, as it originates solely from primary sources. The formation of particle


phase chloride have secondary sources from chlorine atom-initiated oxidation of volatile organic
compounds, so that the resulting oxidation products could contribute to the observed chloride (Wang
and Ruiz, 2017; Wang et al., 2019a).

To better illustrate the combined effects of secondary aerosol formation and associated feedback on
the daytime mass concentrations and light extinction due to different chemical components, we scaled
these quantities by either the total $PM_{2.5}$ mass concentration or EC concentration and plotted them as
a function of MLH (Fig. 7). The latter scaling minimizes the boundary layer accumulation effect on
our analysis, as EC originates from primary emission sources (Cao et al., 2006). As shown in Fig. 7a,
organics with their mass fraction of 61% were the most abundant component in $PM_{2.5}$ under high
MLH conditions, followed by nitrate and ammonium with their mass fractions of 22% and 13%,
respectively. The aerosol was estimated to be rather dry under high MLH conditions ($AWC/PM_{2.5}$ =
0.03). However, with the decreasing MLH, the fraction of nitrate and the AWC to $PM_{2.5}$ ratio increased
up to 45% and 0.2, respectively. This clearly indicates rapid nitrate formation and dramatic increase
of the aerosol water uptake from less polluted conditions to intensive haze pollution. Compared with
EC (Fig.7c), the concentrations of organic compounds, nitrate, sulfate and ammonium increased by
factors of 1.5, 6.3, 4.8 and 4.9 respectively, from the highest to the lowest MLH conditions. Thus,
although organics remained as the second most abundant aerosol component after nitrate under haze
conditions, secondary formation and associated feedback from less to highly polluted conditions were
clearly stronger for both sulfate and ammonium. Efficient sulfate production associated with haze
formation has been reported in several studies conducted in China (Cheng et al., 2016; Xie et al.,
2015; Xue et al., 2016). Ammonium production during haze formation is tied with neutralization of
acidic aerosol by ammonia, which was apparently present abundantly in the gas phase. Compared
with the EC concentration, light extinction by ($NH_4NO_3$) increased the most from the highest MLH
conditions (248 M $m^{-1}$/µg $m^{-3}$) to the lowest MLH conditions (1150 M $m^{-1}$/µg $m^{-3}$) as shown by Figure
7b. Overall, the rapid growth of nitrate aerosol mass, together with abundant concentration of organic
aerosol, were the main cause of the light extinction for dry aerosol under haze formation.



The mechanism governing the aerosol-chemistry-radiation-boundary layer feedback for the rapid
growth of atmospheric aerosol is illustrated in Fig. 8. As a result of reduction in solar radiation and
atmospheric heating, a variety of chemical reactions in the gas phase and on particle surfaces or inside
them are enhanced with an increased relative humidity and AWC. Such conditions are unfavorable
for the dispersion of pollutants, which further enhances atmospheric stability. The formation of
hydrophilic compounds, e.g., nitrate, sulfate and oxygenated organic compounds, result in enhanced
water uptake by aerosol particles, which will essentially increase heterogeneous reactions associated
with these particles. As a result, the aerosol mass and size increase, light extinction is enhanced, and
the development of the mixing layer is depressed. At the same time, aerosol precursors concentrated
within a shallower mixing layer lead to enhanced production rate of aerosol components in both gas
and aerosol phases, especially nitrate but also other secondary aerosol. The increased concentrations
of aerosol will further enhance this positive loop.

**4.  Conclusions**

We investigated the synergetic variations of aerosol chemical composition and mixing layer height
during the daytime in urban Beijing. Significant dependency of the sharp increase of ammonium
nitrate and aerosol water content with the occurrence of the explosive aerosol mass growth events
were observed. We showed that these two components drove a positive aerosol-chemistry-radiation-
boundary layer feedback loop, which played an important role in the explosive aerosol mass growth
events. A plausible explanation is that the increased aerosol water content at low mixing layer heights
provides favorable conditions for heterogeneous reactions for nitrate and sulfate production and
neutralization by ammonia. The significant formation of secondary aerosol increases the
concentration of aerosol particles in the diameter range 300-700 nm, which effectively reduces the
solar radiation reaching the surface and further enhances the aerosol-chemistry-radiation-boundary
layer feedback loop. Our analysis connects the aerosol light extinction to a reduction in the mixing
layer height, which suppresses the volume into which air pollutants are emitted and leads to an



explosive aerosol mass growth. Our results indicate that reduction of ammonium and nitrate
concentration in aerosol could weaken the aerosol-radiation-chemistry-boundary layer feedback loop,
which could thereby reduce heavy haze episodes in Beijing.

## 5. Acknowledgements

This work was supported by the funding from Beijing University of Chemical Technology. The
European Research Council via advanced grant ATM-GTP (project no. 742206) and Academy of
Finland via Academy professor project of M. K.

## 6. Competing financial interests

The authors declare no competing financial interests.

## 7. Author contributions

YW and MK initiated the study. ZL, YW, FZ, YZ, YG, ZF, CL, YZ, TC, CY, KD, BC, JK, LY, XF,
WD, JC and YL conducted the longtime measurements. ZL, YW, LD, RC, SH, PZ, LW, VK, YL and
MK interpreted the data. ZL, YW and VK wrote the manuscript.






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



















Figure caption




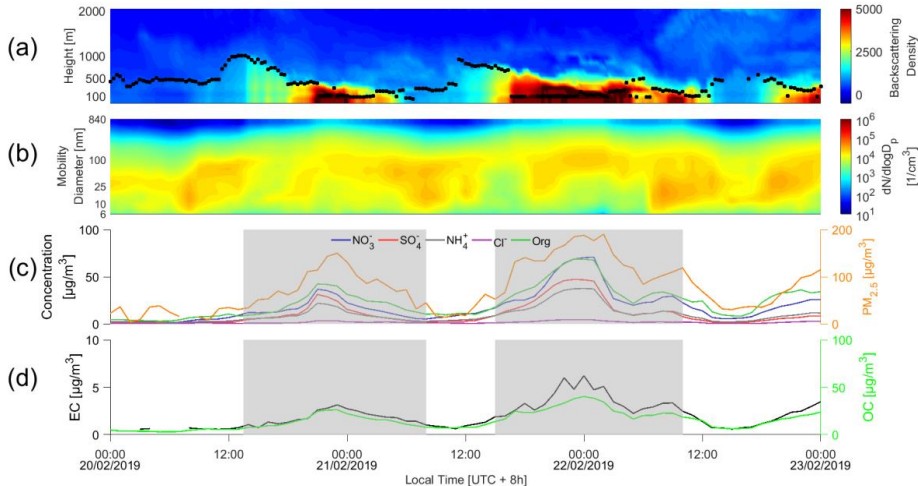



Figure 1. Time series of (a) backscattering density and boundary layer height (b) particle number
concentration distribution (PNSD), (c) chemical composition species mass concentration and PM$_{2.5}$
mass concentrations and (d) elemental carbon (EC) and organic carbon (OC). The haze periods are
marked by the shaded areas.














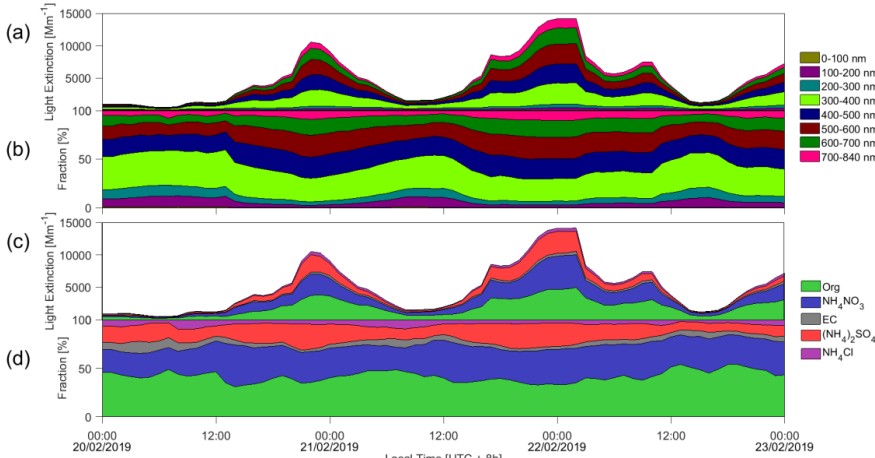


Figure 2. Time series of (a, b) variation of light extinction from different size aerosol and fractions,
and (c, d) variation of light extinction from different aerosol species and fractions. The legends in the
left side of figures are particle diameter and chemical compositions, respectively.















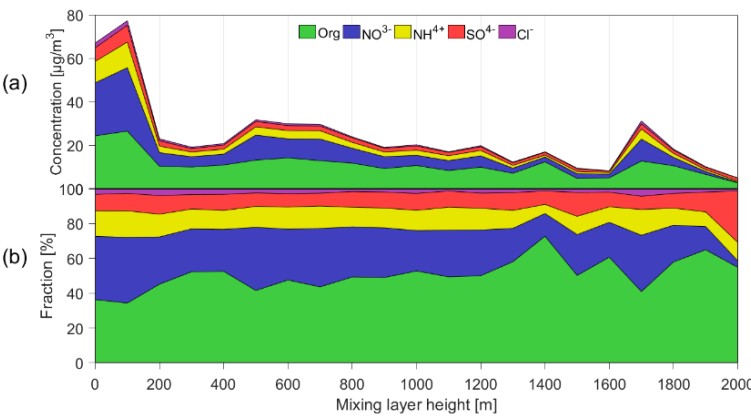


Figure 3. Statistical relationship between MLH and concentration (a) and fraction (b) of chemical composition species. Only daytime conditions determined by ceilometer from non-rainy periods (RH<95%) are considered.


















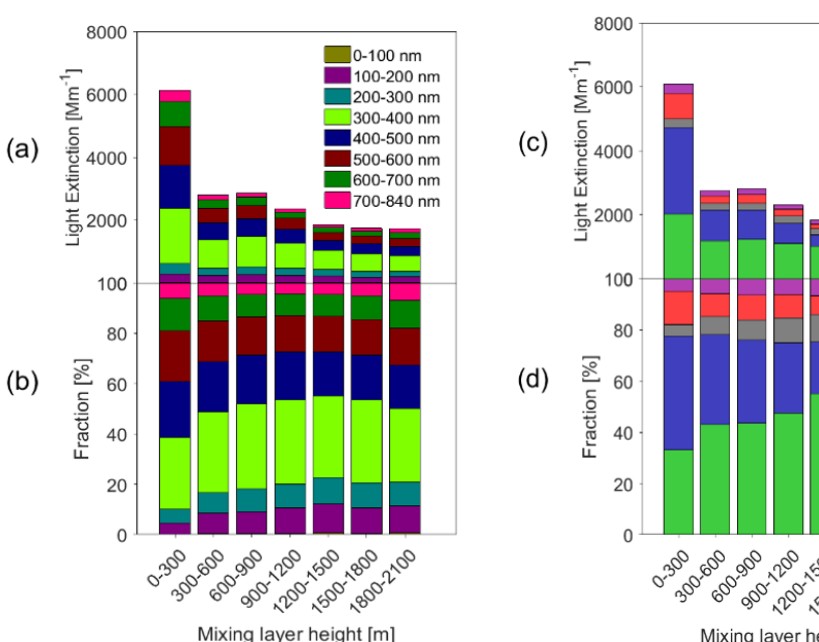

Figure 4. Statistical relationship between MLH and light extinction of different aerosol species. Only daytime conditions determined by the ceilometer from non-rainy periods (RH<95%) are considered.



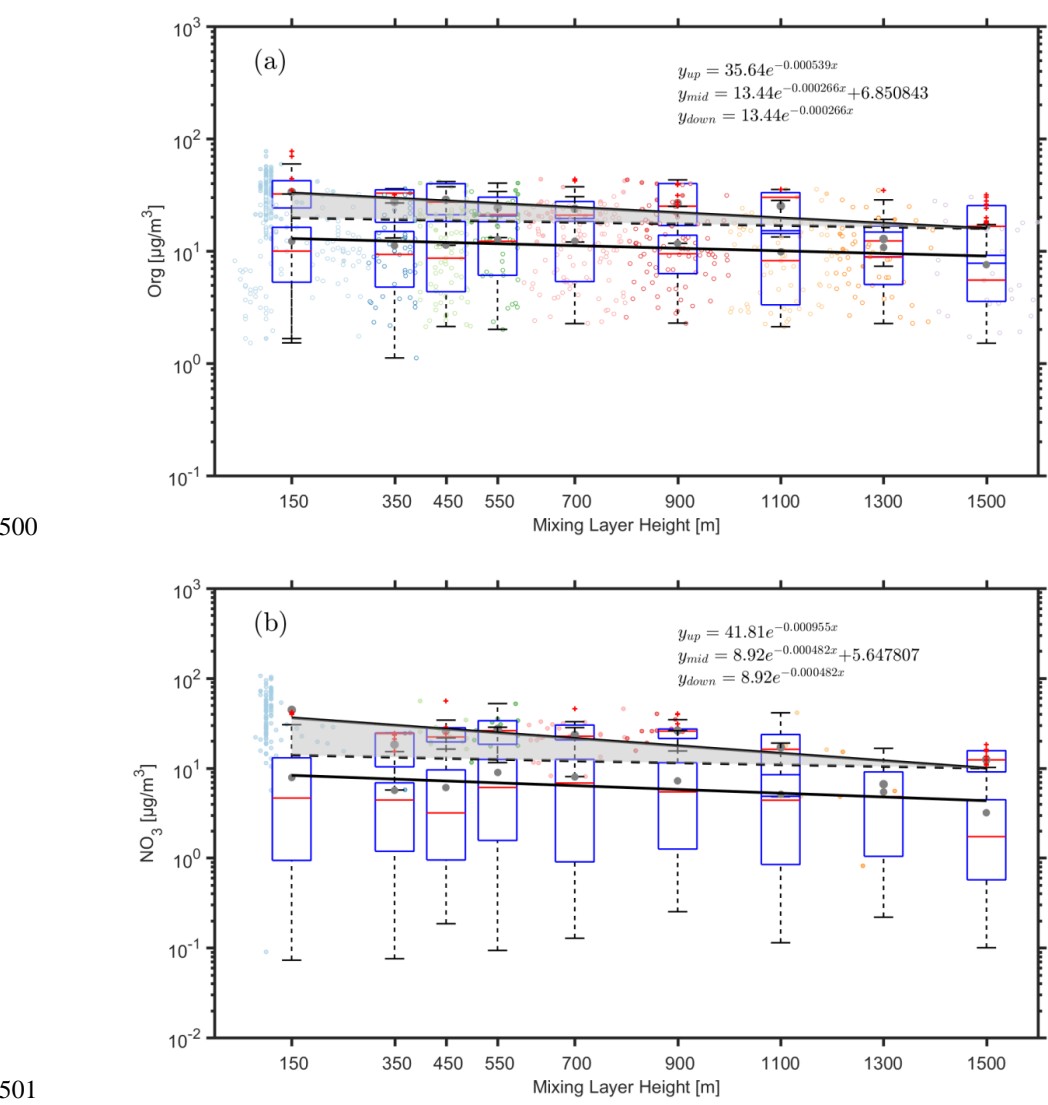





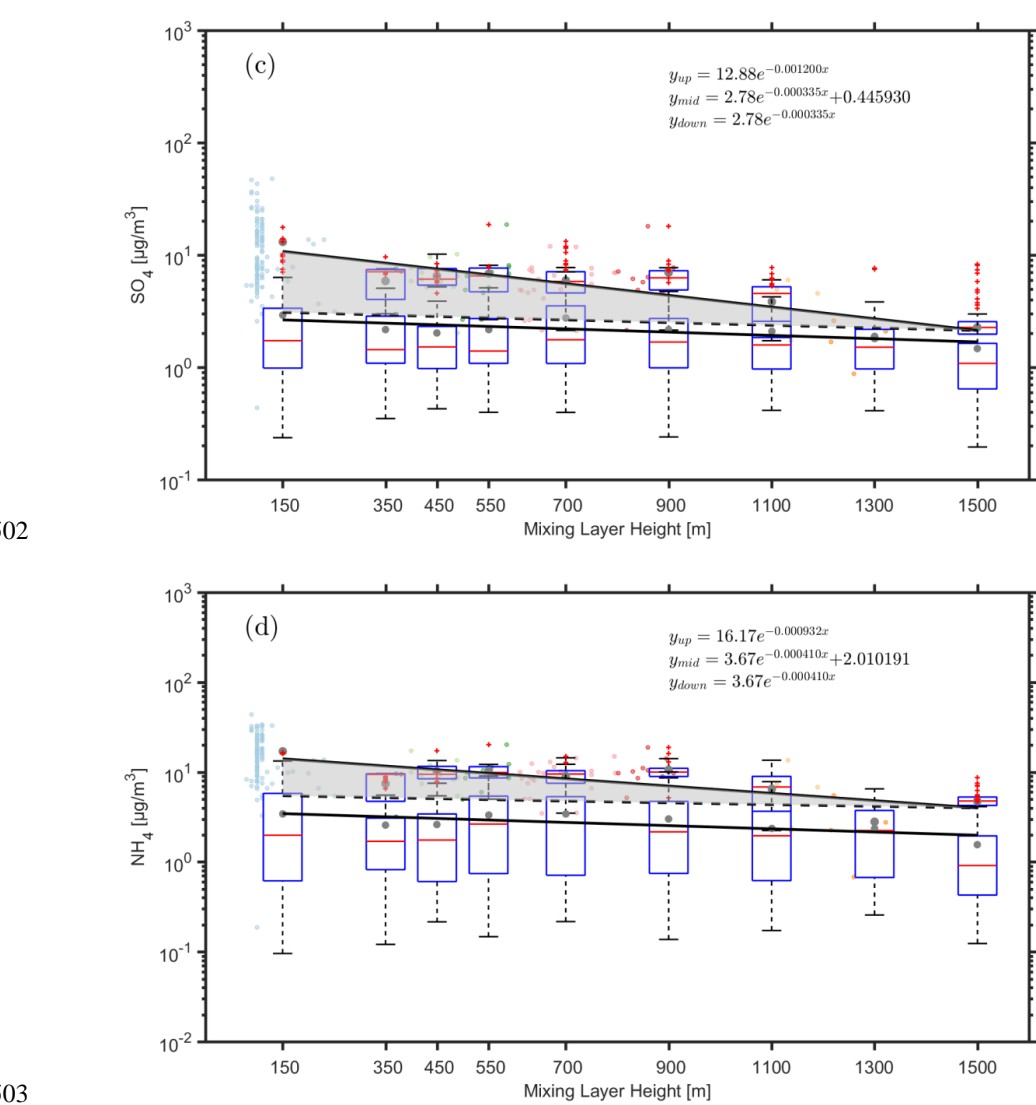







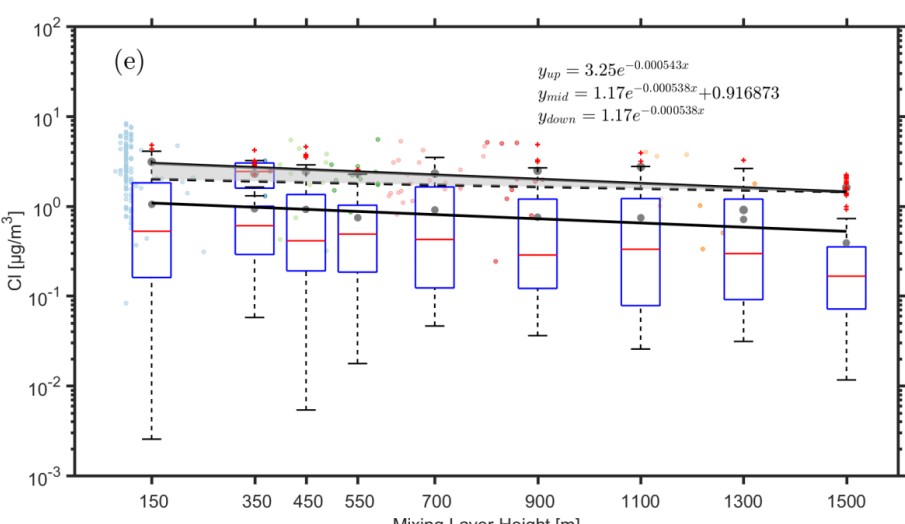




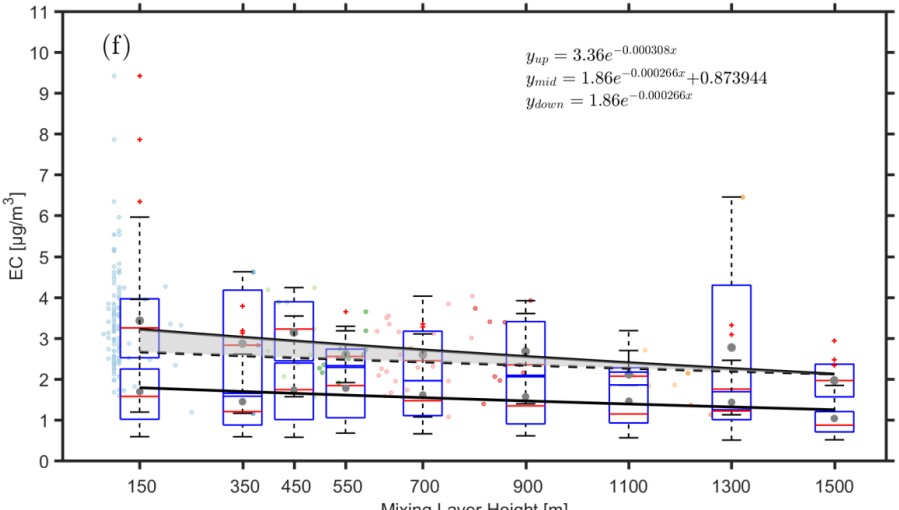




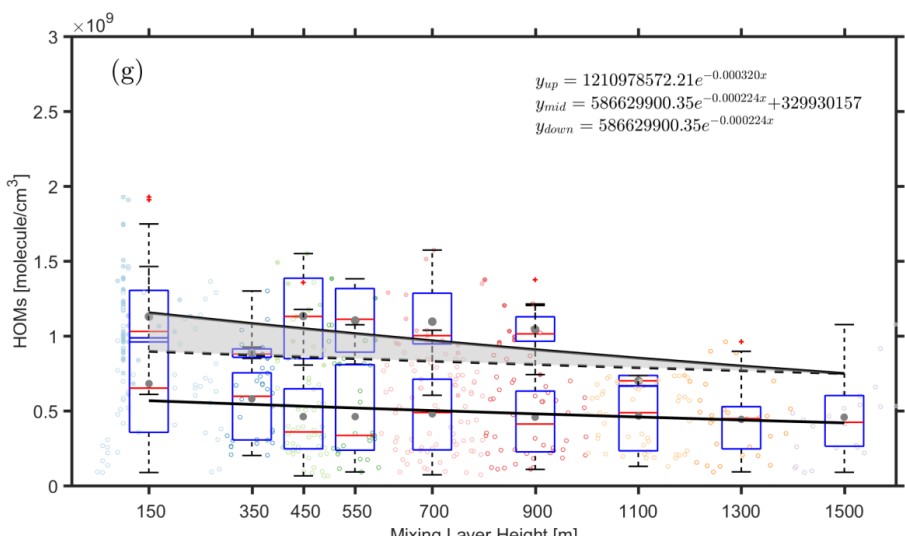


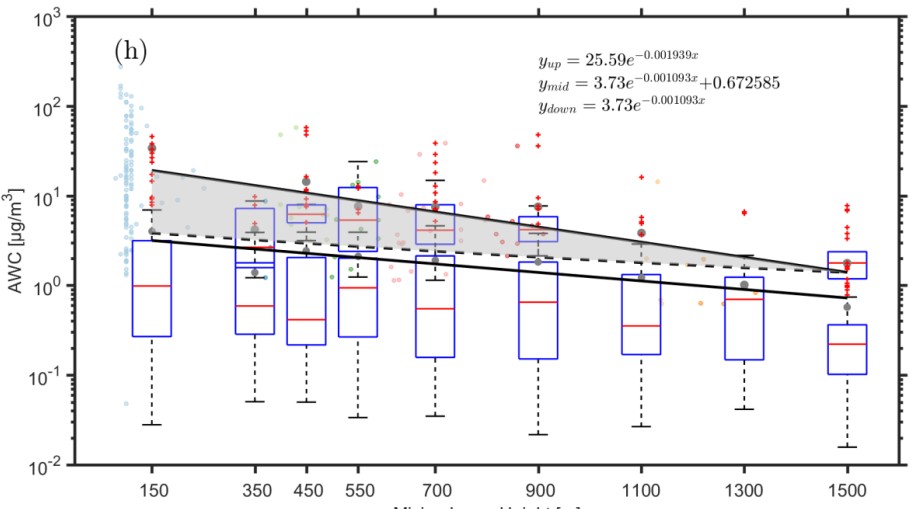




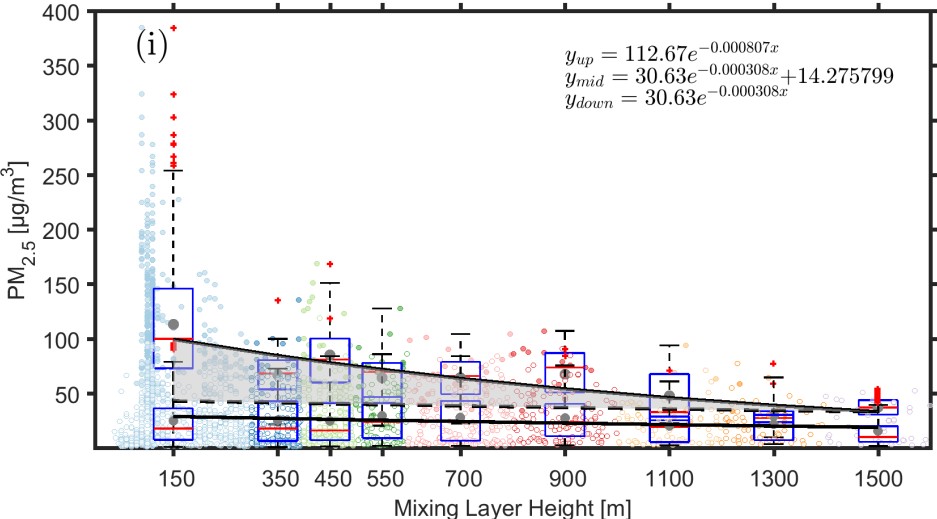


Figure 5. Observed dependency of (organics (a), nitrate (b), ammonium (c), sulfate (d), chlorine (e),

element carbon (f), HOMs (g), AWC (h) and PM$_{2.5}$(i) on the MLH during polluted and less-polluted

conditions. The data related to the upper fitting line represents PM$_{2.5}$ concentrations larger than 75 µg

m$^{-3}$, while the date related to the lower fitting line represents PM$_{2.5}$ concentrations lower than 75 µg

m$^{-3}$. Only daytime conditions determined by the ceilometer from non-rainy periods (RH<95%) were

considered. The dark grey points and red lines in the boxes represent mean and median values,

respectively. The shaded area between the upper solid and dotted lines corresponds to an increased

amount of the specific compounds with decreased MLH, assuming that the compound has the same

variation pattern under highly- polluted conditions as in less polluted time.












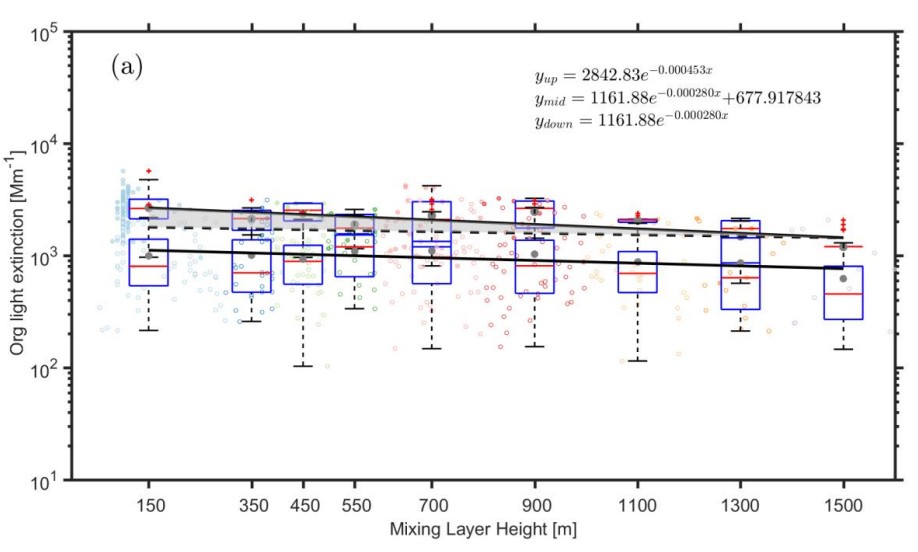


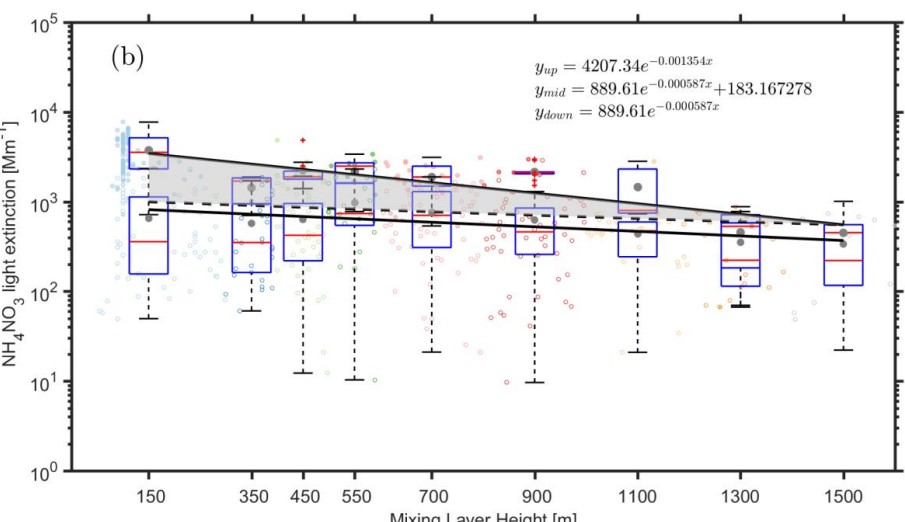






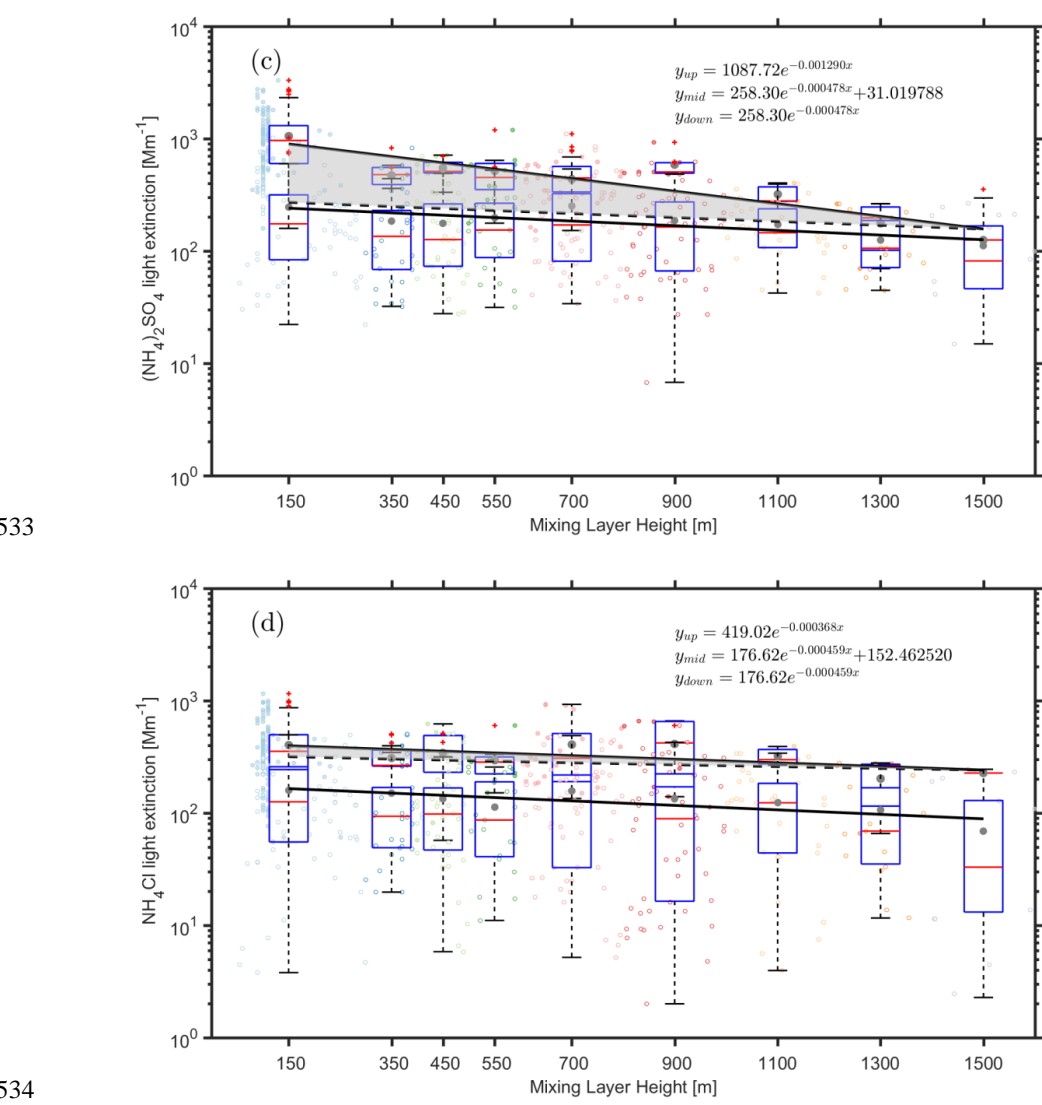



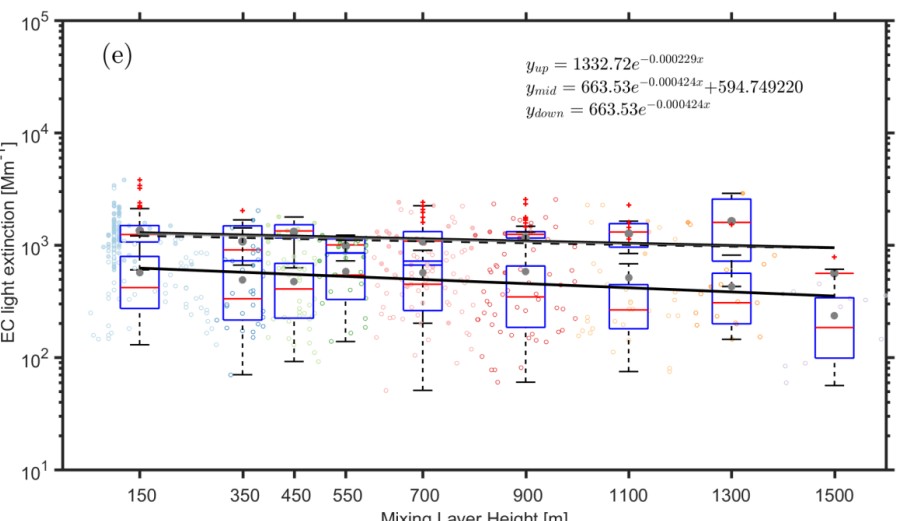




Figure 6. Observed dependency of the aerosol light extinction due to $NH_4NO_3$ (a) $(NH_4)_2SO_4$ (b),
$NH_4Cl$ (c) Org (d) and EC (e) on the MLH during polluted and non-polluted conditions. The data
related to the upper fitting line represents $PM_{2.5}$ concentrations larger than 75 μg m$^{-3}$, while the date
related to the lower fitting line represents $PM_{2.5}$ concentrations less than 75 μg m$^{-3}$. Only daytime
conditions determined by ceilometer from non-rainy periods (RH<95%) are considered. The dark
grey points and red lines in the boxes represent mean and median values, respectively. The shaded
area between the upper solid and dashed line corresponds to an increased amount of $PM_{2.5}$ with a
decreased MLH, assuming that $PM_{2.5}$ has the same variation pattern under highly- polluted conditions
as in less polluted time














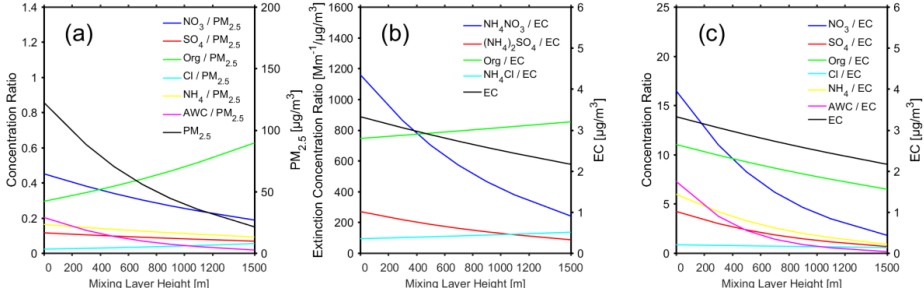


Figure 7. (a) the ratio of the mass concentration of different chemical components (nitrate, sulfate,
organics, chlorine, ammonium) and AWC to the mass concentration of NR_PM$_{2.5}$ as a function of
MLH. (b) the ratio of dry aerosol light extinction by different chemical components (NH$_4$NO$_3$,
(NH$_4$)$_2$SO$_4$, Org, NH$_4$Cl) to the mass concentration EC as a function of MLH (c) the ratio of the mass
concentration of different chemical components (nitrate, sulfate, organics, chlorine, ammonium) and
AWC to the mass concentration of EC as a function of MLH. All the date corresponds to polluted
conditions (fine PM >75 μg m$^{-3}$), and only daytime conditions determined by the ceilometer from
non-rainy periods (RH<95%) were considered.


















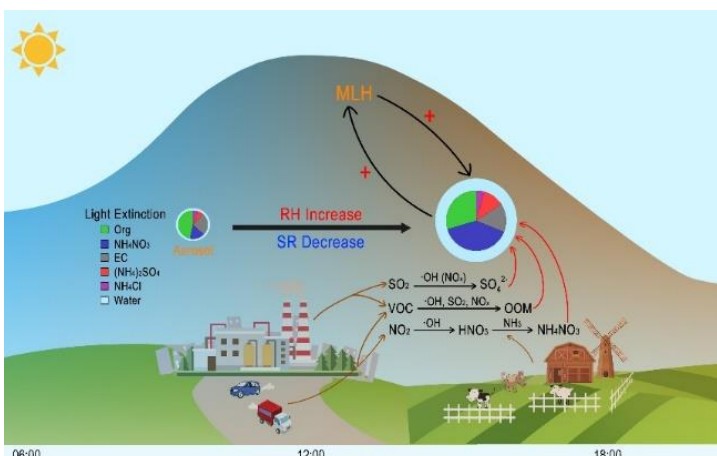


Figure 8. **A schematic picture illustrating the process of rapid aerosol mass growth and enhanced**
**light extinction in Beijing**. The plus symbols represent the strengthening of a specific process. At
the presence of aerosols during afternoon time in Beijing, the intensity of solar radiation reaching the
surface will be decreased and relative humidity will be increased. As a result, the development of
boundary layer will be suppressed, and the concentrations of aerosol precursors (e.g., $SO_2$, $NO_2$, VOC)
will be increased. In turn, the secondary production of these sulfate, nitrate and oxygenated organic
compounds will be enhanced due to increased concentrations and partitioning of these compounds
into the aerosol phase. The increased formation of secondary aerosol mass will reduce solar radiation
further and the haze formation increased. Noting that during intensive haze periods, nitrate and its
contribution to light extinction contribution increased dramatically.



