# Peer review of "Rapid mass growth and enhanced light extinction of atmospheric aerosols during the heating"

_Atmospheric Chemistry and Physics, 2020_

## Referee Comment (RC1) · Anonymous Referee #1 · 14 Jun 2020

This manuscript is to investigate the dependency of the aerosol number size distribution, mass concentration and chemical composition on the daytime mixing layer height (MLH) in urban Beijing. The valuable measurement datasets, especially for oxygenated organic molecules (HOMs), are firstly showed during heating time in China, according to my knowledge. These results show that the haze pollution is rapidly formed by aerosol-chemistry-radiation feedback, which is an interesting topic. By using measured aerosol chemical composition and Mie calculation of light extinction, they reached a

conclusion that ammonium nitrate was the dominated compound under lowest MLH. The conclusion is reasonable considering large amount of on-road vehicles and previous publications. Generally, the results in this manuscript are useful to support policy-makers on air pollution controls in the future. Also, this manuscript is easy to follow and the figures are presented in proper forms. Nevertheless, several statements are needed to be clarified. I suggest this paper could be published after minor revisions as below.

Minor comments:

1. Please clarify the differences between mixing layer height (term used in this study) and boundary layer height.

2. This work is mainly focused on particle number size distribution measurements. A Particle Sizer Magnifier (PSM) and a Differential Mobility Particle Sizer (DMPS) is used in the measurement, so this reviewer is wondering how is the variation of particle number size distribution looks like under different mixing layer height condition under haze and non-haze days? You have already showed how are the response of aerosol chemical component with different mixing layer height. This kind of analysis may tell us particle growth under haze and non-haze period.

3. In your schematic picture, you show haze evolution with the daily mixing layer height. Light extinction of dry aerosol is also assigned to different chemical compounds, however, this information was not mentioned in figure caption, please explain more on these two pie charts in the figure caption.

4. In page4 line 83, "...particles with diameters of a few hundred nm", a given range of the diameters with the constant will be better, if possible, please give them; please add the references for this sentence.

5. In page9 line 221: "... increase from few ug/m3", please change the "few" to specific value.

6. In page10 line 249: if the "NH4NO3" is appeared first, please give the full name.

7. In page10 line 249: if the "EC" is appeared first, please give the full name, also please check for NH4Cl.

8. In page11 lines 277-279: the English grammar tense is inconsistent in the sentence of "We may see that in general, particles with dry diameters in the range of 300-700 nm explained more than 80% of the total aerosol light extinction (Figure 4b)."

9. In page 12 lines 303: the units of "ug m-3" is inconsistent with that of "ug/m3" in page 9 line 222, make sure they are consistent in the full manuscript.

10. Figure 1 caption: explain what PM2.5 represents.

11. Figure 2 caption: "The legends in the left side . . .", it is "right"?

12. Figure 3 caption: please explain the range for the "daytime conditions".

―――――――――――――――――――

---

## Referee Comment (RC2) · Anonymous Referee #3 · 23 Jun 2020

**Review of Lin et al., "Rapid mass growth and enhanced light extinction of atmospheric aerosols during the heating season haze episodes in Beijing revealed by aerosol-chemistry-radiation-boundary layer interaction".**

This paper describes measurements of aerosol physical and chemical properties that were collected over a five month period in Beijing city. These data are combined with boundary layer height measurements, and calculated aerosol scattering efficiency to better understand the aerosol-radiation-boundary feedback relationship. The authors focus on a case study period of three days in February. The results illustrate that as boundary layer decreases and as pollution conditions develop, the contribution of nitrate containing particles increase dramatically. It is suggested that these larger ammonium nitrate particles contribute most towards the reduction in visibility during haze events and also slow the development of the mixing layer height. The low mixing layer height together with the increased concentration of particles is thought to favor the formation of secondary nitrate, sulfate and organic aerosols, and hence leads to marked increases in aerosol particle concentrations. This manuscript presents an interesting data set, although it feels that the authors are just skimming the surface of what could potentially be a very relevant study. Prior to publication, it is necessary to include additional details on instrument operation, and case study periods as outlined in the comments below.

**General comments**

There are many details of instrument operation, including that of the ACSM and also the LTOFMS that are omitted from the paper and supplementary material. The authors focus on measurements for the period from October 2018 to February 2019, but later on only shown data for a three day period in February. Please state this clearly in the abstract, introduction, and methods section if this is the only haze event encountered during this five month sampling period? If not how representative is this haze event compared to other events.

**Introduction:**

Please highlight better the added value of this work compared to previous studies (cited in the references) on the aerosol-radiation-boundary layer feedback.

**Methods:**

1. No information is provided on the inlet set up? How is the aerosol dried prior to sampling?
2. A ToF-ACSM fitted with a PM 2.5 inlet was used in this study. This is still a relatively new version of the instrument, and it merits a correct introduction. Please state if this instrument operating with a standard vaporizer or a capture vaporizer? .The paper referenced here "Frohlich et al., "deployed a PM1 inlet and not a PM2.5 inlet. Please update the references.
3. What collection efficiency was applied to this data? Please show a plot of how the total mass measured by the ACSM compared with that of the TEOM (also PM2.5), how representative is the PM2.5 ACSM measurements of the total PM2.5.
4. Line 132: The authors state that they applied the correction for the m/z 44 artefact, without showing if this instrument was influenced by this artefact, please provide the artefact values calculated from this instrument from pure ammonium nitrate calibrations. More recent studies (Freney et al., AST 2019)

have additionally shown that this correction is often only necessary at highest $NO_3$ mass fractions (> 40%). From Figures 2 through to 4, these contributions never appeared to be greater than 40%.

5. Please also state in the text the average values as well as the range for each species measured (for the period that concerns this study).

6. Additionally data from a LTOF-CIMS is provided. This is a complex instrument, and both the operation and the analysis of this data require a considerable amount of work. Please provide more details on the operation of this instrument and the subsequent analysis of the data. Unlike the ACSM used in this study, this instrument is usually operated with a PM1 inlet rather than PM2.5.

7. The authors need to provide comparisons of measurements between the ACSM and the LTOFCIMS. Please provide additional details for this instrument. Was it sampling alongside the ACSM for similar sampling periods?

8. For the OC/EC measurements, Can the authors also provide plots comparing the OM from the sunset with that of the ToF-ACSM and to the LTOFCIMS. How do the O/C plots compare with that of the LTOFCIMS and calculated from the ACSM with that measured by the OC of the Sunset instrument.

**Results and Discussion**

1. Figure 5 shows data collected during high and low pollution events. Are the differences between the high and low pollution periods significant for all measured species? The authors could perform a significance test (e.g. Wilcoxon rank-sum test).

2. Can the authors provide an estimation of how good these fits represent the data? Can these fits be used in the future to estimate the variability of the aerosol concentration over pollution/haze events?

3. In each plot there are data points (behind the box plots) that are different colors can the authors please provide an adequate legend for this figure.

4. Line 272: When comparing the NH4 neutralization plots were there any periods where neutralization was not achieved that would suggest the presence of organic nitrate. The LTOFCIMS instrument is capable of providing a good assessment of the presence of organic nitrates.

5. Line 304: This is the first mention of the results of HOMS, can the authors also provide time series of these data together with those of the ACSM.

6. Line 315: It is mentioned that there is abundant ammonia but not mentioned if it is measured here. However in Fig. S4 there are plots of NH3 as a function of MLH. How were these measurements obtained?

7. Line 320 and Figure 5. It appears in this figure that during a high pollution event the increase in SO4 is more significant than nitrate.

8. Equally the HOMS appear to have a greater increase during polluted events compared with organics who have a little increase. Previously, it is mentioned that the OA decreases with low MLH. The authors should provide a detailed discussion of HOMS and OA. Also with a simple positive matrix factorization analysis of the ACSM data it would be possible to obtain additional information on the different "types" of organic aerosol measured. In Figure S4 we observe the OC increasing in a similar way to the HOMS. This could also be discussed.

9. Were there any gas phase measurements available to help in the interpretation of the formation of NO3?

**Supplementary material**

There are 12 plots (3 lots of four labeled images a) through d)) in Figure S2. These plots are not sufficiently and incorrectly described in the figure caption, which refers to "different times" please indicate the times and labels (a) to (e). There is no 'e'.

Figure S3 starts at b) rather than a).

Can the authors improve the caption explanation of the figure. These emission sensitivities represent polluted /periods of high aerosol loadings. The figures show high values coming initially from the west and also from the south. In the figures there is little contribution from the north east sectors.

Figure S4: Only one panel is labelled with "f)". The others nothing. There are two representations of the sub 3 nm clusters (-dN/DlogDp and cm/1). Are both necessary? What is the difference between the two?

Figure S4: I do not believe that any reference is made to this plot in the main text of the manuscript, nor to any moeasurements of OH, HONO, NH3 [ppb]
* * *
**Minor comments**

**Abstract:** There is a repetition of information. Line 37 to 39 states that as the MLH decrease the fraction of nitrate aerosol and the total mass concentration increases. This is stated again in Line 41 where the ammonium nitrate and aerosol water increased during low MLH.

**Table 1:** Species instead of Specie

**Line 180:** Please include the reference to the previous publication here.

**Line 231:** I don't believe that this acronym was correctly defined (NR-PM2.5)

**Line 249**:Can the authors rephrase tis sentence: Assuming that particles of different sizes have the same chemical composition as PM2.5 (organics, NH4NO3, EC, (NH4)2SO4, NH4Cl), the light extinction of particles in the size range of 300-700 nm increased significantly from the relative clean period to the polluted period (namely from 12:00 to 16:00).
**Line 255:** Based on the available data, it might be better to say "that based on the observations it is likely that….
**Line 305 to 309:** This is a very long sentence, please try to rephrase.

**Line 346:** remove "rather" this is

**Line 358**: Remove brackets around AN.

Also add in 'the calculated' light extinction.

The authors mention the presence of a PSM but no measurements are shown. The SMPS data start at 6 nm

**Figure 3:** Since all other figures are based on 3 days of analysis please state in the figure caption the period that this data is collected over .In the main text it is suggested that this figure represents 6 months of data.

In the caption text change "All the date" to "All the data"

---

## Author Comment (AC1) · 11 Nov 2020

The comment was uploaded in the form of a supplement:
https://acp.copernicus.org/preprints/acp-2020-223/acp-2020-223-AC1-supplement.pdf

---

## Author Comment (AC2) · 11 Nov 2020

A point to point response to the reviewers' comments

We thank the two reviewers for their comments, and we think their comments and suggestions improved our manuscript. Here are points to points responses (in blue colored), accordingly, we also revised manuscript (in blue colored).

Reviewer #1

General comments: There are many details of instrument operation, including that of the ACSM and also the LTOFMS that are omitted from the paper and supplementary material. The authors focus on measurements for the period from October 2018 to February 2019, but later on only shown data for a three day period in February. Please state this clearly in the abstract, introduction, and methods section if this is the only haze event encountered during this five month sampling period? If not how representative is this haze event compared to other events.

Response: At first we considered the data of these instruments as supporting proof materials, but later we thought that it is very likely that the details of these instruments need to be explained. We will introduce these instruments in the subsequent comments to prevent repetition.

Actually, it's not the only one haze event during this five month sampling period. We chose a dozen of haze events based on the our measurements and this haze event shown in Figure 1 is a typical event. As the MLH decreases, with high relative humidity and a sharp rise in the concentration of various pollutants, the mass of particulate matter has shown explosive growth. This is a good example to describe that Haze has been suggested to be initiated by the variation of meteorological

parameters and then to be substantially enhanced by aerosol-radiation-boundary layer feedback.

Introduction: Please highlight better the added value of this work compared to previous studies (cited in the references) on the aerosol-radiation-boundary layer feedback.

Response: Previous studies are mainly focused on physical mechanism of aerosol-radiation-boundary layer interaction, which is a case of rapid haze formation in China. However, The chemistry of aerosol composition that influence the interaction loop is not studied yet, the novelty of the work compared with previous ones are well demonstrated in introduction section. Line: 92-95.

Methods: 1. No information is provided on the inlet set up? How is the aerosol dried prior to sampling?

Response: The ToF-ACSM equipped with a $PM_{2.5}$ lens and standard vaporizer. A $PM_{2.5}$ cyclone was deployed on the rooftop with a flow rate of 3 L/ min. Aerosol was dried though a Nafion dryer (MD-700-24F-3, PERMA PURE) before entering the ToF-ACSM. The inlet flow was set at 1.4 cc/s. We added these introduction in revised version. Line: 126-128.

2. A ToF-ACSM fitted with a PM 2.5 inlet was used in this study. This is still a relatively new version of the instrument, and it merits a correct introduction. Please

state if this instrument operating with a standard vaporizer or a capture vaporizer? .The paper referenced here "Frohlich et al., "deployed a PM1 inlet and not a PM2.5 inlet. Please update the references.

Response: ToF-ACSM method: The time-of-flight aerosol chemical speciation monitor (ToF-ACSM, Aerodyne Research Inc.) is used to measure the concentrations of non-refractory (NR) components, including sulfate, nitrate, ammonium, chloride and organics. The ACSM equipped with a PM2.5 lens and standard vaporizer. A PM2.5 cyclone was deployed on the rooftop with a flow rate of 3 L min-1. Aerosol was dried though a Nafion dryer (MD-700-24F-3, PERMA PURE) before entering the ACSM. The inlet flow was set at 1.4 cc/s. The particle beam passed through the chamber and reaches the heated porous tungsten surface (T600). There the non-refractory PM2.5 constituents vaporized and were ionized by electrons (Ekin=70eV, emitted by a tungsten filament). The ions were measured by detector and the data was analyzed using the software (Tofware ver. 2.5.13) within IgorPro ver. 6.3.7.2 (Wavemetrics). The relative ionization efficiencies (RIE) for sulfate, nitrate, ammonium, chloride and organics applied were 0.86, 1.05, 4.0, 1.5 and 1.4, respectively. Except RIE correction, the data also did $CO_2+$/ $NO_3$ artifact correction (Pieber et al., 2016) and collection efficiency (CE) correction (Middlebrook et al., 2012).

3. What collection efficiency was applied to this data? Please show a plot of how the total mass measured by the ACSM compared with that of the TEOM (also PM2.5),

how representative is the PM2.5 ACSM measurements of the total PM2.5.

Response: The sampled aerosol belongs to High Ammonium Nitrate Fraction during the sampling period of Beijing. Therefore the CE correction following by these equations (Middlebrook et al., 2012):

$$ANMF = \frac{80/62 \times NO_3}{(NH_4 + SO_4 + NO_3 + Chl + Org)}$$

where $NH_4$, $SO_4$, $NO_3$, Chl and Org were the measured aerosol ammonium, sulfate, nitrate, chloride, and organic concentrations (in $\mu g \cdot m^{-3}$).

$$CE_{dry} = max(0.45, 0.0833 + 0.9167 \times ANMF)$$

in which a constant CE of 0.45 is used for ANMF ≤ 0.4 and a linear CE increase up to 1 for ANMF > 0.4. We compared measurements of ACSM with TEOM as in figure below.

[Figure]

Figure R1. The relationship between $PM_{2.5}$ measured by TEOM and ToF-ACSM.

4. Line 132: The authors state that they applied the correction for the m/z 44 artefact, without showing if this instrument was influenced by this artefact, please provide the artefact values calculated from this instrument from pure ammonium nitrate

calibrations. More recent studies (Freney et al., AST 2019)

Response: Recently, it was discovered that NO3 induces a positive bias on organic $CO_2+$ concentrations in the AMS/ACSM systems, which can be described as a function of ambient NO3 (µg/m3) in combination with the CO2+/NO3 ratio from pure NH4NO3 measurements (CO2+/NO3)AN:

For pure NH4NO3 aerosol from calibrations, we determined the magnitude of the CO2+/NO3 artifact(Pieber et al., 2016) and parametrized it as a function of the fragmentation pattern of NO3(NO+/NO2+) to account for changes in the vaporizer in the ACSM:

$$(CO_2^+/NO_3)_{NH4NO3} = 0.025 \pm 0.002 \times (NO^+/NO_2^+)_{NH4NO3}$$

Then we determined the CO2 concentration from OA using a two week moving average (NO+/NO2+) from ambient observations:

$$(CO_2^+)_{OA,meas} = (CO_2^+)_{meas} - (CO_2^+/NO_3)_{NH4NO3} \times (NO_3)_{meas}$$

5. Please also state in the text the average values as well as the range for each species measured (for the period that concerns this study).

Response: Thanks for your suggestion. We also added the table in supporting information.

| Species | Mean | Min | Max |
|---|---|---|---|
| AWC [µg/m^3] | 10.1972 | 0.0157 | 279.9762 |
| NH3 [ppb] | 8.1979 | 1.4475 | 24.2622 |
| HOMs | 6.4557*10^8 | 6.5261*10^7 | 2.7647*10^9 |

| [molecule/cm^3] | | | |
|---|---|---|---|
| HONO [ppb] | 1.3799 | 0.1352 | 10.4820 |
| EC [μg/m^3] | 2.4212 | 0.5025 | 19.9765 |
| OC [μg/m^3] | 11.0719 | 1.4217 | 114.3976 |
| OH | 5.3011*10^5 | 236.2 | 5.2052*10^6 |
| NO3 [μg/m^3] | 15.7131 | 0.0310 | 126.8300 |
| SO4 [μg/m^3] | 5.5307 | 0.1951 | 117.5360 |
| NH4 [μg/m^3] | 6.4492 | 0.0913 | 51.4603 |
| Cl [μg/m^3] | 1.6346 | 0.0025 | 17.0581 |
| Org [μg/m^3] | 19.1311 | 0.6662 | 172.8490 |
| $PM_{2.5}$ [μg/m^3] | 50.8277 | 0.1592 | 218.5980 |

6. Additionally data from a LTOF-CIMS is provided. This is a complex instrument, and both the operation and the analysis of this data require a considerable amount of work. Please provide more details on the operation of this instrument and the subsequent analysis of the data. Unlike the ACSM used in this study, this instrument is usually operated with a $PM_1$ inlet rather than $PM_{2.5}$.

Response: The LTOF-CIMS instrument is operated with nitrate as reagent ion. We used total OVOC concentration calculated from calibration of sulfuric acid. The instrument did not use any inlet cyclones. Nitrate chemical ionization atmospheric pressure interface time-of-flight (CI-APi-TOF, Aerodyne Research, Inc.) mass

spectrometers were used to measure the concentrations of neutral sulfuric acid and HOMs. The ambient air was drawn into the ionization source through a stainless-steel tube with a length of ~1.6 m and a diameter of 3/4 inch at a flowrate of ~ 8 L·min-1. A 30-40 L·min-1 purified air flow and a 4-8 mL·min-1 ultrahigh purity nitrogen flow containing nitric acid were mixed together as the sheath flow, which is guided through a PhotoIonizer (Model L9491, Hamamatsu, Japan) to produce nitrate reagent ions. This sheath flow is then introduced into a co-axial laminar flow reactor concentric to the sample flow. Nitrate ions are pushed to the sample flow layer by an electric field and subsequently charge analytical molecules.

The calibration of sulfuric acid (SA) was implemented by introducing a known amount of gaseous SA produced by the reaction of SO2 and OH radical formed by UV photolysis of water vapor, which is similar to the method in previous literatures (Andreas Kürten et al., 2012). Briefly, a 10 L·min-1 N2 flow, a 100 mL·min-1 purified air flow, a 300 mL·min-1 SO2 flow and a set of 20 – 400 mL·min-1 saturated water vapor flow were mixed together as the calibration sampling flow. This flow was introduced into the calibration box where the water vapor was photolysed by a 184.9 nm UV light and the producing OH radicals further reacted with SO2 to form SA. Different concentrations of SA standards were achieved by adjusting the flow of saturated water vapor. During the calibration, the box was flushed with a 1 – 2 L·min-1 dry N2 flow to avoid the absorption of UV light by O2 and water vapor as well as take off the heat produced by the lamp. This N2 flow was directly fed into the box through a small hole and left it through the small gaps between different parts.

Besides, the UV lamp was always turned on in an N2 environment at least one hour before the actual calibration measurement in order to achieve a stable light intensity. Theoretical concentrations of SA at the inlet were simulated by a numerical tube model (Andreas Kürten et al., 2012). And the calibration coefficient was further calculated from the ratio between the theoretical concentration and the normalized ion intensity. After taking the diffusion loss of the sampling line into account, a calibration coefficient of $6.07 \times 10$-9 molecule·cm-3 was obtained.

As the structures of these newly detected HOMs are unknown, direct calibration of using HOM standard is impossible yet. By assuming that HOM charge at their collision frequency with nitrate ions, which is the case for H2SO4 (Viggiano, A. A. et al., 1997), and that the (HOM·NO3-) clusters are very stable and will not break apart during their residence time of detection, a mass-dependent transmission method was used to quantify their concentrations. Details of this approach is described elsewhere (Martin Heinritzi et al., 2016). Briefly, for each instrument, the transmission calibration measurements were performed by introducing a series of perfluorinated acid vapors of different molecular masses with sufficient amounts to consume all the primary ions. Then by comparing the decrease of the primary ion signals and the increase of added perfluorinated acid signals, the relative transmission curve was obtained. Such mass-dependency is highly influenced by the configuration and parameters of specific instrument, especially the voltage settings. Besides, some studies have shown that less oxygenated organic molecules with lower polarity exhibit less charged efficiency and weaker bound with NO3- (Martin Breitenlechner

et al., 2017; Noora Hyttinen et al., 2015). Thus, the reported concentration of HOMs in this study is generally a lower limit. And the concentration of each OOM is calculated as follows:

$$[HOM] = \frac{\sum_{i=0}^{1}(HNO_3)_iNO_3^-(HOM) + (HNO_3)_i(HOM-H)^-}{\sum_{i=0}^{2}(HNO_3)_iNO_3^-} \times C \div T_{HOM}$$

where [HOM] is the concentration of one specific HOM molecule, the numerator on the right hand side is the sum of detected signal of that HOM, either as neutral molecule or as de-protonated ion (HOM)-, the denominator is the sum of all measured reagent ions, C is the calibration factor of H2SO4 and total HOM is the relative transmission coefficient. We added more introduction in the revised version. Line: 142-149.

7. The authors need to provide comparisons of measurements between the ACSM and the LTOFCIMS. Please provide additional details for this instrument. Was it sampling alongside the ACSM for similar sampling periods?

Response: The Figure R2 shows time series of HOMs and OA measured by LTOFCIMS and ACSM, respectively. In general, both organic compounds in particle phase and gas phase show similar variation patterns during the same observation periods. The details for the instruments are introduced in response 6.

[Figure]

Figure R2. Time series of HOMs and OA measured by LTOFCIMS and ACSM, respectively.

8. For the OC/EC measurements, Can the authors also provide plots comparing the OM from the sunset with that of the ToF-ACSM and to the LTOFCIMS. How do the O/C plots compare with that of the LTOFCIMS and calculated from the ACSM with that measured by the OC of the Sunset instrument.

Response: The Figure R 3 shows time series of HOMs, OC and OA measured by LTOFCIMS, Sunset OC/EC analyzer and ACSM, respectively. They showed similar variation patterns during the same observation periods.

[Figure]

Figure R3. Time series of HOMs, OC and OA measured by LTOFCIMS Sunset OC/EC analyzer and ACSM, respectively

Results and Discussion 1. Figure 5 shows data collected during high and low pollution events. Are the differences between the high and low pollution periods significant for all measured species? The authors could perform a significance test (e.g. Wilcoxon rank-sum test).

Response: We use the function ranksum of MATLAB to perform Wilcoxon rank-sum test.

p = ranksum(x,y) returns the p-value of a two-sided Wilcoxon rank sum test. ranksum tests the null hypothesis that data in x and y are samples from continuous distributions with equal medians, against the alternative that they are not. The test assumes that the two samples are independent. x and y can have different lengths.

This test is equivalent to a Mann-Whitney U-test.

The result h = 1 indicates a rejection of the null hypothesis, and h = 0 indicates a failure to reject the null hypothesis at the 5% significance level.

Example:

p = ranksum(x,y)

p = 0.0375

The p-value of 0.0375 indicates that ranksum rejects the null hypothesis of equal medians at the default 5% significance level.

| Species | p | h |
|---|---|---|
| AWC [µg/m^3] | 5.4286*10^(-76) | 1 |
| NH3 [ppb] | 1.2178*10^(-55) | 1 |

| | | |
|---|---|---|
| HOMs [molecule/cm^3] | 8.7649*10^(-42) | 1 |
| HONO [ppb] | 2.1083*10^(-29) | 1 |
| EC [μg/m^3] | 2.2462*10^(-61) | 1 |
| OC [μg/m^3] | 2.83*10^(-82) | 1 |
| OH | 6.1802^(-4) | 1 |
| NO3 [μg/m^3] | 1.6328*10^(-91) | 1 |
| SO4 [μg/m^3] | 6.5457*10^(-80) | 1 |
| NH4 [μg/m^3] | 1.2669*10^(-91) | 1 |
| Cl [μg/m^3] | 3.5606*10^(-63) | 1 |
| Org [μg/m^3] | 1.2495*10^(-79) | 1 |
| $PM_{2.5}$ [μg/m^3] | 8.0856*10^(-113) | 1 |

2. Can the authors provide an estimation of how good these fits represent the data? Can these fits be used in the future to estimate the variability of the aerosol concentration over pollution/haze events?

Response: We used a linear fit between the log(x)(Org, $NO_3$, $SO_4$, $NH_4$, Cl, AWC, $NH_4NO_3$ light extinction, $(NH_4)_2SO_4$ light extinction, $NH_4Cl$ light extinction, Org light extinction and EC light extinction) and the MLH or an Exponential fit between x(EC, HOMs, $PM_{2.5}$) and the MLH. This fittings are only validated during observation periods and for other periods, it might work.

3. In each plot there are data points (behind the box plots) that are different colors can the authors please provide an adequate legend for this figure.

Response: The figure show the dependency of (organics (a), nitrate (b), ammonium (c), sulfate (d), chlorine (e), element carbon (f), HOMs (g), AWC (h) and PM$_{2.5}$(i) on the MLH during polluted and less-polluted conditions. **The solid cycles and hollow cycles denotes concentrations that are more than 75 μg m$^{-3}$ and less than 75 μg m$^{-3}$, respectively**. We also added this information in figure caption in the revised version.

4. Line 272: When comparing the NH4 neutralization plots were there any periods where neutralization was not achieved that would suggest the presence of organic nitrate. The LTOFCIMS instrument is capable of providing a good assessment of the presence of organic nitrates.

Response: We agree with the reviewer that organic nitrates as a good assessment indicate NH4 neutralization was not achieved. But NH4 in any periods is neutralized in the NH4 neutralization plots, because the NH4 measured by ACSM can almost be neutralized with the measured Cl, NO3, SO4. Actually, we don't need organic nitrates to prove neutralization.

5. Line 304: This is the first mention of the results of HOMS, can the authors also provide time series of these data together with those of the ACSM.

Response: Thank you for your suggestion. This is the first time show the concentrations of HOM measured by NO3-CI-API-TOF as mentioned by the other reviewer. However, this study is not focused on details of HOM chemistry, the

concentration shown here is higher than a magnitude than the measurements in boreal forest region. The concentrations showed high concentrations during haze event than clean days and increased significantly during night time. Please see previous responses for more information.

6. Line 315: It is mentioned that there is abundant ammonia but not mentioned if it is measured here. However in Fig. S4 there are plots of NH3 as a function of MLH. How were these measurements obtained?

Response: The ammonia data were measured in the same place (at the roof top of the university building at the west campus of Beijing University of Chemical Technology). Los Gatos Research, Inc. (LGR) trace Ammonia analyzer (TAA) can measure NH3 and H2O concentrations at atmospheric ambient levels with high precision (0.2 PPB in 1s) and ultra-fast response (5 Hz). We added the instruments introduction in method part. Line: 151-152.

7. Line 320 and Figure 5. It appears in this figure that during a high pollution event the increase in SO4 is more significant than nitrate.

Response: Yes. The growth of sulfate is comparable with nitrate, or sometimes even fast than the nitrate. However, the concentration of sulfate with nitrate is not comparable. Nitrate shows higher concentration but the sulfate concentration is lower, as we show in Figure 6.

8. Equally the HOMS appear to have a greater increase during polluted events compared with organics who have a little increase. Previously, it is mentioned that the OA decreases with low MLH. The authors should provide a detailed discussion of HOMS and OA. Also with a simple positive matrix factorization analysis of the ACSM data it would be possible to obtain additional information on the different "types" of organic aerosol measured. In Figure S4 we observe the OC increasing in a similar way to the HOMS. This could also be discussed.

Response: From Figure R 3 annd R4, we could see that both HOM and OA showed increased or decreased patterns, as the MLH varies. We have discussion in line 331-334. We acknowledge that the comparison of HOM and OA will be extremely interesting, and we have a another draft (to be submitted) about the relationship of HOM molecules with organic aerosol factors from PMF. As we demonstrated in the introduction part, the current paper is to investigate chemistry of aerosol-boundary layer- radiation feedback, and there are a lot of interesting points waiting to be explored in the future.

9. Were there any gas phase measurements available to help in the interpretation of the formation of NO3?

Response: I think you mean nitrate aerosol not nitrate radical here. The formation of nitrate aerosol The formation of nitrate is dominated by the oxidation of NO2 by hydroxyl radical (OH) during the daytime but the heterogeneous reaction of N2O5 during the nighttime. We do not have measurements of N2O5 and gas phase nitric acid during the campaign.

Supplementary material There are 12 plots (3 lots of four labeled images a) through d)) in Figure S2. These plots are not sufficiently and incorrectly described in the figure caption, which refers to "different times" please indicate the times and labels (a) to (e). There is no 'e). Figure S3 starts at b) rather than a). Can the authors improve the caption explanation of the figure. These emission sensitivities represent polluted /periods of high aerosol loadings. The figures show high values coming initially from the west and also from the south. In the figures there is little contribution from the north east sectors. Figure S4: Only one panel is labelled with "f)". The others nothing. There are two representations of the sub 3 nm clusters (-dN/DlogDp and cm/1). Are both necessary? What is the difference between the two? Figure S4: I do not believe that any reference is made to this plot in the main text of the manuscript, nor to any moeasurements of OH, HONO, NH3 [ppb]

Response: The figure captions have been revised. The two representations of the sub 3 nm clusters (-dN/DlogDp and cm/1) denote particles with different size and particles with size under 3 nm, the (-dN/DlogDp) one has been removed in the revised version. Also, the Figure S4 has been removed.

Minor comments Abstract: There is a repetition of information. Line 37 to 39 states that as the MLH decrease the fraction of nitrate aerosol and the total mass concentration increases. This is stated again in Line 41 where the ammonium nitrate and aerosol water increased during low MLH.

Response: We mainly want to show that the main component of nitrate aerosols is

ammonium nitrate, after all, nitrate aerosols are not completely ammonium nitrate.

Table 1: Species instead of Specie

Response: corrected.

Line 180: Please include the reference to the previous publication here.

Response: We added references in the revised version. Line: 189.

Reference : Lin, Z. J., Tao, J., Chai, F. H., Fan, S. J., Yue, J. H., Zhu, L. H., Ho, K. F. and Zhang, R. J.: Impact of relative humidity and particles number size distribution on aerosol light extinction in the urban area of Guangzhou, Atmospheric Chemistry and Physics, 13(3), 1115–1128, doi:10.5194/acp-13-1115-2013, 2013.

Line 231: I don't believe that this acronym was correctly defined (NR-PM2.5).

Response: We added fully name in the revised version. Line: 240.

Line 249: Can the authors rephrase this sentence: Assuming that particles of different sizes have the same chemical composition as PM2.5 (organics, NH4NO3, EC, (NH4)2SO4, NH4Cl), the light extinction of particles in the size range of 300-700 nm increased significantly from the relative clean period to the polluted period (namely from 12:00 to 16:00).

Response: The light extinction efficiency of aerosol is highly dependent on aerosol chemical composition. However, size-resolved chemical information from ToF-ACSM is not available due to the instrument limitation. Therein, we assume that particles of different sizes have the same chemical composition as $PM_{2.5}$ measured in the study  (organics, NH4NO3, EC, (NH4)2SO4, NH4Cl) in the light extinction

calculation.

Line 255: Based on the available data, it might be better to say "that based on the observations it is likely that….

Response: corrected in the revised version. Line: 263

Line 305 to 309: This is a very long sentence, please try to rephrase.

Response:

Line 346: remove "rather" this is

Response: corrected.

Line 358: Remove brackets around AN. Also add in 'the calculated' light extinction.

Response:

The authors mention the presence of a PSM but no measurements are shown. The SMPS data start at 6 nm.

Response: We added two plots about variation of sub 3 nm particle with MLH as shown in supporting information and the responses to reviewer #2. The SMPS data starts at 6 nm and the data has been demonstrated.

Figure 3: Since all other figures are based on 3 days of analysis please state in the figure caption the period that this data is collected over .In the main text it is suggested that this figure represents 6 months of data. In the caption text change "All the date" to "All the data"

Response: The Figure 3 is based not only 3 days of data as we demonstrated in Figure caption, it was from 6 months of calculated from non-rainy days. The figure caption has been corrected.

Referencnes

Middlebrook, A.M., Bahreini, R., Jimenez, J.L., and Canagaratna, M.R. (2012). Evaluation of Composition-Dependent Collection Efficiencies for the Aerodyne Aerosol Mass Spectrometer using Field Data. Aerosol Science and Technology 46, 258-271.

Pieber, S.M., El Haddad, I., Slowik, J.G., Canagaratna, M.R., Jayne, J.T., Platt, S.M., Bozzetti, C., Daellenbach, K.R., Frohlich, R., Vlachou, A., et al. (2016). Inorganic Salt Interference on CO2(+) in Aerodyne AMS and ACSM Organic Aerosol Composition Studies. Environ Sci Technol 50, 10494-10503.

Andreas Kürten, Linda Rondo et al., Calibration of a chemical ionization mass spectrometer for the measurement of gaseous sulfuric acid. Journal of Physical Chemistry A, 2012.

Viggiano, A. A., Seeley, J. V. et al., Rate constants for the reactions of XO3-(H2O)n (X = C, HC, and N) and NO3-(HNO3)n with H2SO4: implications for atmospheric detection of H2SO4. Journal of Physical Chemistry A, 1997.

Martin Heinritzi, Mario Simon et al., Characterization of the mass-dependent transmission efficiency of a CIMS. Atmospheric Measurement Techniques, 2016.

Martin Breitenlechner, Lukas Fischer et al., PTR3: An Instrument for Studying the Lifecycle of Reactive Organic Carbon in the Atmosphere. Analytical Chemistry, 2017.

Noora Hyttinen, Oona Kupiainen-Määtta et al., Modeling the Charging of Highly Oxidized Cyclohexene Ozonolysis Products Using Nitrate-Based Chemical Ionization, Journal of Physical Chemistry A ,2015.

Referrer #2

This manuscript is to investigate the dependency of the aerosol number size distribution, mass concentration and chemical composition on the daytime mixing layer height (MLH) in urban Beijing. The valuable measurement datasets, especially for oxygenated organic molecules (HOMs), are firstly showed during heating time in China, according to my knowledge. These results show that the haze pollution is rapidly formed by aerosol-chemistry-radiation feedback, which is an interesting topic. By using measured aerosol chemical composition and Mie calculation of light extinction, they reached aconclusion that ammonium nitrate was the dominated compound under lowest MLH. The conclusion is reasonable considering large amount of on-road vehicles and previous publications. Generally, the results in this manuscript are useful to support policymakers on air pollution controls in the future. Also, this manuscript is easy to follow and the figures are presented in proper forms. Nevertheless, several statements are needed to be clarified. I suggest this paper could be published after minor revisions as below.

Response: Thank you for your positive comments.

Minor comments:

1. Please clarify the differences between mixing layer height (term used in this study) and boundary layer height.

   Response: According to the definition by Holzworth 1972: Mixing layer height is defined as the height above the surface through which relatively vigorous vertical mixing occurs. But BL is more generally defined as part of the troposphere that is directly influenced by the presence of the earth's surface. We measured vertical backscattering coefficient by CL-51, and determined MLH according to the variation of backscattering coefficient.

2. This work is mainly focused on particle number size distribution measurements. A Particle Sizer Magnifier (PSM) and a Differential Mobility Particle Sizer (DMPS) is used in the measurement, so this reviewer is wondering how is the variation of particle number size distribution looks like under different mixing layer height condition under haze and non-haze days? You have already showed how are the response of aerosol chemical component with different mixing layer height. This kind of analysis may tell us particle growth under haze and non-haze period.

Response: The variation of particle diameter with MLH under haze and non-haze days are shown in Figure S3. The black line in (up) is the location of 50% of the total particle number concentration (PM2.5 >= 75 μg/m3). The black line in (down) and The dotted line in (a) are the location of 50% of the total particle number concentration (PM2.5 < 75 μg/m3). Only daytime conditions determined by ceilometer CL51 from non-rainy periods (RH<95%) are considered. The particle show a slight diameter increasing as MLH increased above 400 meters. Particles shows larger mean diameters during polluted periods than non-polluted periods.

[Figure]

Figure S4 The relationship between MLH and PNSD in (a) polluted and (b) non-polluted days. The black line in (a) is the location of 50% of the total particle number concentration (PM2.5 >= 75 μg/m3). The black line in (b) and The dotted line in (a) are the location of mean diameter (PM2.5 < 75 μg/m3). Only daytime conditions determined by ceilometer CL51 from non-rainy periods (RH<95%) are considered.

[Figure]

3. In your schematic picture, you show haze evolution with the daily mixing layer height. Light extinction of dry aerosol is also assigned to different chemical compounds, however, this information was not mentioned in figure caption, please explain more on these two pie charts in the figure caption.

Response: Thank you for your suggestion. We have added more explanation about the pie charts in the figures caption. " The increased formation of secondary aerosol mass will reduce solar radiation further and the haze formation increased, as shown in pie

charts that the light extinction fraction of aerosol changed from organic to nitrate."

4. In page4 line 83, ". . .particles with diameters of a few hundred nm", a given range of the diameters with the constant will be better, if possible, please give them; please add the references for this sentence.

Response: The sentence has been revised as: 'In the atmosphere, the highest contribution to aerosol light extinction comes from organic compounds, nitrate and sulphate in particles with diameters of 100-1000 nm.'

5. In page9 line 221: ". . . increase from few ug/m3", please change the "few" to specific value.

Response: The sentence has been revised as ' 8.5 ug/m3'.

6. In page10 line 249: if the "NH4NO3" is appeared first, please give the full name.

Response: The full name has been added in line 134.

8. In page10 line 249: if the "EC" is appeared first, please give the full name, also please check for NH4Cl.

Response: The full name of EC and others have been explained in Line 142 and Line 134.

8. In page11 lines 277-279: the English grammar tense is inconsistent in the sentence of "We may see that in general, particles with dry diameters in the range of 300-700 nm explained more than 80% of the total aerosol light extinction (Figure 4b)."

Response: The sentence has been revised as' We may see that in general, particles

with dry diameters in the range of 300-700 nm explains more than 80% of the total aerosol light extinction (Figure 4b)'

9. In page 12 lines 303: the units of "ug m-3" is inconsistent with that of "ug/m3" in page 9 line 222, make sure they are consistent in the full manuscript.

Response: The units are consistent in the revised version.

10. Figure 1 caption: explain what PM2.5 represents.

Response: The PM2.5 represents particles with aerodynamic diameter less than 2500 nm, we think this terminology is familiar with the readers.

11. Figure 2 caption: "The legends in the left side . . .", it is "right"?

Response: Thank you for the correction, we have revised this.

12. Figure 3 caption: please explain the range for the "daytime conditions".

Response: The daytime conditions is used during the observations with solar radiation.

---

## Referee Report (RR1)

Thank you for taking the time to provide a response and clarifications to the reviewer's comments. However, despite having provided a lot of information and figures, the authors failed to integrate these changes into the manuscript. The last version of the supplementary material dates to November 11, 2020, therefore unless it has been explicitly stated in the response to reviewer's questions it is unclear if more changes have been included.

My suggestions and comments of how this manuscript can be improved are detailed below.

\*\*

**General introduction to the sampling period:** It would be useful to provide an overview section of the measurements into the results section, prior to introducing the case study measurements. The authors can then provide information on why the chosen period is a representative case study. In this section, the long term temporal trend of the ACSM measurements, the OC from Sunset and the HOMS measurements, as well as the size distributions from both measurements (DMPS +PSM), could be included.

**Comparison with collocated measurements:** The authors provided details of the comparison between the NRPM25 ACSM and the PM25 from the TEOM. This comparison should at least be included in the supplementary section on the manuscript and this agreement should be referred to in the main text.

From the figure provided in R1, it appears that there are a small number of points (above the fit line) where the agreement between the ACSM and the TEOM are better. Do these data points correspond to any period in particular (Haze events/nonevent days)? Is this level of agreement TEOM (or other external measurements) considered good for the PM25 inlet of the ACSM, please provide references? Can the authors show this fit as a function of time.

\*\*\*

In general recommendations, it is better to place a PM10 inlet upwind of a PM25 inlet rather than applying twice a PM25 size cut. This can result in an unnecessary loss of particulate mass.

\*\*

**Regarding the CO2 artefact.** Thank you for the response to the reviewers queries, however it would be expected to include these details into the main part (or at least the supplementary part) of the manuscript, and to include information as to how this artefact varied throughout the sampling period.

The correction described by the authors is a modification of that described in Pieber et al., so it is necessary to provide a reference for this modification or at least a more detailed description of this method as well as a justification of its use compared to the previous published correction.

The authors should show the impact of this correction on the total mass concentrations (as well as the F44 vs F43 plots) etc. What are the differences in the fit between the ACSM and TEOM with and without the mz44 artefact correction?

Reword «In addition to the RIE correction, a correction for the mz44 artefact (Pieber et al., 2016) and a collection efficiency correction were applied".

**Comparison of ACSM Org and Sunset OC:** In the response to the reviewers, the authors provided a comparison between Org measured by the ACSM and that from the Sunset, these comparisons could be included in an overview section or at least in the supplementary. Can the authors state if a denuder was used upstream of the sunset instrument and include a reference to what analysis protocol was used for the Sunset OC/EC measurements.

Figure R3/Table in supp: It appears that the OC from Sunset values are consistently lower than Org ACSM values? Please provide some explanation for this? What was the OM to OC ratio applied to the Sunset measurements?

**Significance of increases in concentrations during haze events:** Thank you for providing the information on the significance of the aerosol mass concentration increases during the haze events. However, these additions do not appear to be included in the manuscript. A short sentence in the main part of the manuscript and some information in the supplementary would be informative for the reader.

**Measurements from PSM and LTOFCIMS:** Regarding the PSM measurements, they are only mentioned in the methods section and do not appear to be included anywhere else in the manuscript, (neither in the figures or supplementary figures), or used in the discussion. I would suggest either including more discussion of these measurements, or removing the instrument description.

Similarly, the majority of the discussion and the main conclusion of this manuscript is focused on ACSM (and DMPS) measurements, with little focus on the LTOFCIMS measurements (except in relation to Fig. 5), making their added value to this manuscript unclear. In the response to the reviewers comments the authors provided some interesting comments and figures comparing the temporal evolution of the OC (Sunset), Org(ACSM) and HOMS (LTOFCIMS), but did not include this discussion in the manuscript. These plots together with some discussion could also be included in the overview section of the manuscript

**\*\*Minor remarks\*\*\**

Line 317 75  $\mu$ g /m correct to m3

Figure S1: Please provide a legend in the figure or in the figure caption for the gray shaded area.

Figure S2: It is still not clear what each of these groups of four figures refers to. Please provide the dates (and times) on the figures to distinguish them from each other; also a label on the color scale is needed. Also check the caption, there are only figures labeled (a) through (d).

Figure S3: The axis labels could be improved here, they are very small.

Figure S4: Neither a or b is included.

---

## Editor Decision (ED1)

**Review of paper acp-2020-223: Rapid mass growth and enhanced light extinction of atmospheric aerosols during the heating season haze episodes in Beijing revealed by aerosol-chemistry-radiation-boundary layer by Lin et al.**

Dear author, co-authors,

Having found finally the time to carefully check again the reviews, your response to these reviews and revised version of your paper on analysis of the aerosol-chemistry-radiation-BL feedback, I was triggered to still provide an editor's comment. The reviews were generally positive on the presented analysis although there were also some major issues addressed.

Reviewer #3 (you refer to this reviewer being #1) makes a good point about the issue on the number of haze events and how many of these are considered in your actual analysis; you then indicate that you have selected one specific event that can been deemed being representative for many more events. This should be made very clear already in the abstract as indicated also by the reviewer. You address this in your comment but subsequently you have not acted upon this having modified the text to better describe this essential feature of the methodology.

The next point by reviewer #3 is that it should be better stressed what the novel feature is of your work compared to previous studies on the aerosol-chemistry-radiation-BL feedback. You mention then in your response that other studies mainly focused on the physical component of this feedback whereas your study focusses more on the chemistry component; First of all, your reply is very short and not very concise but also see that you have not really handled this comment not having included a more extensive review of the main findings of previous studies and what distinguishes your work from that previous work. For example, reading the sentence 68/70: " **The increased stability of the boundary layer leads to enhanced air pollution in the mixed layer, which further suppresses the development of boundary layer**" triggers the question how then the air pollution is suppressing BL development; is this through reducing surface radiation (short/longwave), energy balance? Is this through the effect on temperature profile? Or is there also a moisture effect (aerosol-water/evapotranspiration)? Your reply is that the text lines 92-95 make clear what the novel feature is but these lines only state what your paper focuses on.

Again, here at the end of the introduction it seems to be very useful to already mention that you mainly present a detailed analysis of one haze event deemed being representative for the multiple events that occurred in the period "October 2018 to February 2019"

Reviewer #3's comment on ToF-ACSM measurement; Your response is long and has been a puzzle to find out how parts of your response has then also been used in the revision. Could you please provide a more optimally organized response clearly indicating which modifications have been included in the revision!

Further reading through the response file and seeing how reviewers #3 comments are being handled, I realize that for this discussion on instrumental issues I really want to invite once more again reviewer #3 to provide an evaluation of some of these details.

But my overall point is that your response is not well organized. It is not clear which part of the long responses have been finally used to also revise the manuscript. Instead of further completing here a detailed editors review, I have decided to first allow you to provide at this stage an improved revision/response to handle this observation and then to reinvite

reviewer #3 to revaluate your revised ms. Note that I checked the whole revision and replies but restrict my detailed comments up a point where I got really got convinced that continuation of this review process first needs now you to provide an improved revision consistent with an improved response file. Anyhow, below you can find some of the minor issues I already found in carefully checking especially the first part of the revision.

Minor comments:

Line 68: ...near-surface air (Ding et al., 2016

Line 72: "creating more favourable conditions for homogeneous and heterogeneous chemistry on aerosol surfaces or inside them.."

Line 138: "calculated by the thermodynamic equilibrium model ISO..."

---

## Author Response (AR3)

Comments to the Author:

Dear author, co-authors,

Having received a review of your revised version of your paper ACP-2020-233 it has become further clear that, in submitting your revision as well as your response to previous comments on the paper but also additional files such as the supplement, that you have not properly handled the shared feedback. Besides these observations also being explicitly mentioned in this new review, I also noticed that in your latest reply you didn't address my editors comments on a potential mix up of document versions. I have actually considered also for this reasons to reject the paper but having received now again this (constructive) review I give you once more again the opportunity to properly handle all those comments and then resubmit in due time a consistent selection of files including your replies (including an explanation what has been going wrong), a revision as well as a supplement that contain the changes as being addressed in your response letter. Hope that this then really allows to focus on the content features of your ms submitted for publication in ACP.

Laurens Ganzeveld

Dear editor,

Thank you for giving us the opportunity to revise the manuscript.

In the previous revised version, the reviewers' comments and suggestions were mainly focused on technical issues of the measurements and the comparisons between different instruments. However, the story of our manuscript was the chemistry of aerosol-boundary layer-solar radiation feedback. Considering the length of the manuscript, we responded the reviewers' comments only in response letter but we did not put the all changes into our revised manuscript. We really apologize for this.

In this revised version, we addressed all the comments and suggestion raised by the reviewer and the editor. Also, a revised version of manuscript and supplement information were uploaded. Please find the point to point response to the reviewer's comments below, and the revised manuscript according to the comments.

Best regards,

Yonghong Wang

A point to point response to the reviewer's report

Thank you for taking the time to provide a response and clarifications to the reviewer's comments. However, despite having provided a lot of information and figures, the authors failed to integrate these changes into the manuscript. The last version of the supplementary material dates to November 11, 2020, therefore unless it has been explicitly stated in the response to reviewer's questions it is unclear if more changes have been included.

My suggestions and comments of how this manuscript can be improved are detailed below.

We thank the referee for the fruitful comments, and we think these comments and suggestions improved our manuscript. We have made these changes into our manuscript. Here are points to points responses (in blue colored), accordingly, we also revised manuscript (in blue colored).

**General introduction to the sampling period:** It would be useful to provide an overview section of the measurements into the results section, prior to introducing the case study measurements. The authors can then provide information on why the chosen period is a representative case study. In this section, the long term temporal trend of the ACSM measurements, the OC from Sunset and the HOMS measurements, as well as the size distributions from both measurements (DMPS +PSM), could be included.

Response: Thank you for the comments and suggestions, and the overview section of the measurements was added in the manuscript (section 3.1). In that section, long term temporal trend of the ACSM measurements, the OC from Sunset and the HOMS measurements and the size distributions DMPS were plotted and introduced.

[Figure]

Figure R1. Time series of (a) particle number concentration distribution (PNSD) from 6 nm to 840 nm (b) chemical composition of NR_PM$_{2.5}$ and PM$_{2.5}$ mass concentrations (c) The concentrations of organic carbon (OC) and highly oxygenated organic molecules (HOMs).

**Comparison with collocated measurements:** The authors provided details of the comparison between the NR-PM$_{2.5}$ ACSM and the PM$_{2.5}$ from the TEOM. This comparison should at least be included in the supplementary section on the manuscript and this agreement should be referred to in the main text.

Response: We thank you for the suggestion. The comparison between the NR_PM2.5 ACSM and PM2.5 from TEOM has been added in the supplementary section and the agreement has been referred in the main text (Line: 130-131).

From the figure provided in R1, it appears that there are a small number of points (above the fit line) where the agreement between the ACSM and the TEOM are better. Do these data points correspond to any period in particular (Haze events/nonevent days)?

Response: These data points above the fit line are correspond to haze period in our research (October 1, 2018 ~ February 28, 2019). The points (above the fit line) indicate more higher ratio of NR-PM$_{2.5}$ to PM$_{2.5}$ and higher ratio of inorganic matter to NR-PM$_{2.5}$.

Is this level of agreement TEOM (or other external measurements) considered good for the PM$_{2.5}$ inlet of the ACSM, please provide references? Can the authors show this fit as a function of time.

Response: The two instruments' inlet separate $PM_{2.5}$ using the same method, which ensure constant air flow to get enough accuracy of separated particle size. The maintenance will be performed periodically. We think this the agreement is considered good for the $PM_{2.5}$ inlet of the ACSM[2-5]. Figure R2 shows the fit as a function of time.

[Figure]

Figure R2. Time series of $PM_{2.5}$ mass concentrations and NR-$PM_{2.5}$ mass concentrations.

**Regarding the $CO_2$ artefact.** Thank you for the response to the reviewers queries, however it would be expected to include these details into the main part (or at least the supplementary part) of the manuscript, and to include information as to how this artefact varied throughout the sampling period.

Response: Thank you for the comment. These details have been added in the supplementary part.

**$CO_2^+$/ $NO_3$ artefact correction of ACSM** as follows:

Recently, it was discovered that $NO_3$ induces a positive bias on organic $CO_2^+$ concentrations in the AMS/ACSM systems, which can be described as a function of ambient $NO_3$ ($\mu g/m^3$) in combination with the $CO_2^+/NO_3$ ratio from pure $NH_4NO_3$ measurements $(CO_2^+/NO_3)_{AN}$:

For pure $NH_4NO_3$ aerosol from calibrations, we determined the magnitude of the $CO_2^+/NO_3$ artefact [6] and parametrized it as a function of the fragmentation pattern of $NO_3(NO^+/NO_2^+)$ to account for changes in the vaporizer in the ACSM:

$$(CO_2^+/NO_3)_{NH4NO3} = 0.025 \pm 0.002 \times (NO^+/NO_2^+)_{NH4NO3}$$

Then we determined the $CO_2$ concentration from OA using a two week moving average $(NO^+/NO_2^+)$ from ambient observations:

$$(CO_2^+)_{OA,meas} = (CO_2^+)_{meas} - (CO_2^+/NO_3)_{NH4NO3} \times (NO_3)_{meas}$$

The correction described by the authors is a modification of that described in Pieber et al., so it is necessary to provide a reference for this modification or at least a more detailed description of this method as well as a justification of its use compared to the previous published correction.

Response: The Detailed information of the method can be found in Cai et al. (2020) [1].

The authors should show the impact of this correction on the total mass concentrations (as well as the F44 vs F43 plots) etc. What are the differences in the fit between the ACSM and TEOM with and without the mz44 artefact correction? Reword «In addition to the RIE correction, a correction for the mz44 artefact (Pieber et al., 2016) and a collection efficiency correction were applied".

Response: Response: The resulting $CO_2^+$ artefact bias derived from ACSM measurement should be trivial for Total mass concentration in our research (Figure 1). In the process of nitrate calibration, a signal of m44 was found, which we thought was a false signal from the instrument measurement. In the real measurement process, the effect of 44 should be deducted, in other words, the part of false signal should be subtracted from the organic matter (Cai et al., 2020;Pieber et al., 2016). Actually, we have already considered mz44 artefact correction in OA concentration calculation, where OA signs minus the mz44 signs as correction. The total mass concentrations after correction slightly below the total mass concentrations without correction.

[Figure]

Figure R3. Time series of m44 artefact divided by OA without $CO_2$ correction and m44 artefact divided by OA with $CO_2$ correction (a), m44 artefact divided by total mass with $CO_2$ correction (b).

**Comparison of ACSM Org and Sunset OC:** In the response to the reviewers, the authors provided a comparison between Org measured by the ACSM and that from the Sunset, these comparisons could be included in an overview section or at least in the supplementary.

Response: Thank you for the suggestion. These comparisons have been added in the overview section.

[Figure]

Figure R4. Time series of ACSM Org and Sunset OC.

Can the authors state if a denuder was used upstream of the sunset instrument and include a reference

to what analysis protocol was used for the Sunset OC/EC measurements.

Response: We use a denuder upstream of the sunset instrument. And the analysis protocol called NIOSH-5040[7, 8] was used for the Sunset OC/EC measurements.

Figure R3/Table in supp: It appears that the OC from Sunset values are consistently lower than Org ACSM values? Please provide some explanation for this? What was the OM to OC ratio applied to the Sunset measurements?

Response: OC comes from the carbon part of Org, so OC concentration should be less than Org. In the early 1990s, Sunset Laboratory began to make commercially available thermal-optical OC-EC lab instruments, the Lab OCEC Aerosol Analyzer. In 2000, Model-4 Semi-Continuous OC/EC Field Analyzer, a semi-continuous OC-EC analyzer was developed for near real-time, in-situ measurement of carbon aerosol. However, the instruments from Sunset can't output OM, so that we can get the OM to OC ratio. The average OM/OC ratio is estimated to 1.54 (±0.20 standard deviation) [9] in urban area.

**Significance of increases in concentrations during haze events:** Thank you for providing the information on the significance of the aerosol mass concentration increases during the haze events. However, these additions do not appear to be included in the manuscript. A short sentence in the main part of the manuscript and some information in the supplementary would be informative for the reader.

Response: Thank you for the suggestion. We added a sentence in the revised manuscript. Line: 325.

**Measurements from PSM and LTOFCIMS:** Regarding the PSM measurements, they are only mentioned in the methods section and do not appear to be included anywhere else in the manuscript, (neither in the figures or supplementary figures), or used in the discussion. I would suggest either including more discussion of these measurements, or removing the instrument description.

Similarly, the majority of the discussion and the main conclusion of this manuscript is focused on ACSM (and DMPS) measurements, with little focus on the LTOFCIMS measurements (except in relation to Fig. 5), making their added value to this manuscript unclear. In the response to the reviewers comments the authors provided some interesting comments and figures comparing the

temporal evolution of the OC (Sunset), Org (ACSM) and HOMS (LTOFCIMS), but did not include this discussion in the manuscript. These plots together with some discussion could also be included in the overview section of the manuscript.

Response: Thank you for the suggestion. The PSM introduction in the instrument part has been removed in the revised manuscript. The discussion about measurements of LTOFCIMS have been added in the overview section (3.1 overview of the measurement).

**Minor remarks***

Line 317 75 μg /m correct to m3

Response: corrected.

Figure S1: Please provide a legend in the figure or in the figure caption for the gray shaded area.

Response: The gray shaded area corresponded to period with haze pollution. The figure caption has been refined.

Figure S2: It is still not clear what each of these groups of four figures refers to. Please provide the dates (and times) on the figures to distinguish them from each other; also a label on the color scale is needed. Also check the caption, there are only figures labeled (a) through (d).

Response: Thank you for the suggestion. The figure has been revised according to your suggestion. The dates and the label on the color scale have been added.

Figure S3: The axis labels could be improved here, they are very small.

Response: The axis labels have been improved in the revised version.

Figure S4: Neither a or b is included.

Response: corrected.

[1]  CAI J, CHU B, YAO L, et al. Size-segregated particle number and mass concentrations from different emission sources in urban Beijing [J]. Atmospheric Chemistry and Physics, 2020, 20(21):

12721-40.

[2] CRENN V, SCIARE J, CROTEAU P L, et al. ACTRIS ACSM intercomparison – Part 1: Reproducibility of concentration and fragment results from 13 individual Quadrupole Aerosol Chemical Speciation Monitors (Q-ACSM) and consistency with co-located instruments [J]. Atmospheric Measurement Techniques, 2015, 8(12): 5063-87.

[3] POULAIN L, SPINDLER G, GRüNER A, et al. Multi-year ACSM measurements at the central European research station Melpitz (Germany) – Part 1: Instrument robustness, quality assurance, and impact of upper size cutoff diameter [J]. Atmospheric Measurement Techniques, 2020, 13(9): 4973-94.

[4] FRENEY E, ZHANG Y, CROTEAU P, et al. The second ACTRIS inter-comparison (2016) for Aerosol Chemical Speciation Monitors (ACSM): Calibration protocols and instrument performance evaluations [J]. Aerosol Science and Technology, 2019, 53(7): 830-42.

[5] PETIT J E, FAVEZ O, SCIARE J, et al. Two years of near real-time chemical composition of submicron aerosols in the region of Paris using an Aerosol Chemical Speciation Monitor (ACSM) and a multi-wavelength Aethalometer [J]. Atmospheric Chemistry and Physics, 2015, 15(6): 2985-3005.

[6] PIEBER S M, EL HADDAD I, SLOWIK J G, et al. Inorganic Salt Interference on CO2(+) in Aerodyne AMS and ACSM Organic Aerosol Composition Studies [J]. Environ Sci Technol, 2016, 50(19): 10494-503.

[7] KEVIN ASHLEY P D A P F O C, NIOSH. NIOSH Manual of Analytical Methods (NMAM), 5th Edition [M]. DEPARTMENT OF HEALTH AND HUMAN SERVICES Centers for Disease Control and Prevention National Institute for Occupational Safety and Health, 2016.

[8] BAUER J J, YU X Y, CARY R, et al. Characterization of the sunset semi-continuous carbon aerosol analyzer [J]. J Air Waste Manag Assoc, 2009, 59(7): 826-33.

[9] BROWN S G, LEE T, ROBERTS P T, et al. Variations in the OM/OC ratio of urban organic aerosol next to a major roadway [J]. J Air Waste Manag Assoc, 2013, 63(12): 1422-33.

---

## Author Response (AR4)

Comments to the Author:

Dear author, co-authors, after a review process that has taken a long time, I can finally accept your manuscript on chemistry-aerosol-BL dynamics interactions submitted for publication in ACP. I received a final review and have pasted here the last shared comments that I still want you to handle in providing the final version of the ms for publication in ACP:

"Thank you also for including the information on the $CO_2$+/$NO_3$ artefact. The authors cited in the response to the reviewer that details are provided in the publication Cai et al 2020, can this reference be included in your manuscript and specifically state that the information is detailed in the supplementary material. I have no further comments on this manuscript and I recommend it for publication."

Regards, Laurens Ganzeveld

Dear editor,

Thank you for your inputs on our manuscript.

The revised manuscript has been revised according to review's comment. We want to thank you and the two reviewers for your time and efforts in handling and reviewing this paper.

"Thank you also for including the information on the $CO_2^+/NO_3$ artefact. The authors cited in the response to the reviewer that details are provided in the publication Cai et al 2020, can this reference be included in your manuscript and specifically state that the information is detailed in the supplementary material. I have no further comments on this manuscript and I recommend it for publication."

Response: The reference has been added in the reference list and a sentence has been also added (Line: 140) as the reviewer suggested.

Best regards,

Yonghong Wang